# Attention Hijacking: Backdooring Text Dataset Distillation via Semantic Anchors

**Hang Ren** [1]  **Xin Wang** [1]  **Tong Yue** [1]  **Wen Chen** [1]  **Junqing Le** [2][3]

## Abstract

Dataset Distillation has emerged as a promising technique for compressing large-scale datasets into compact synthetic sets while preserving model performance. However, the security implications of this paradigm, particularly within the Transformer-based text classification domain, remain underexplored. In this paper, we identify "Distilled Attention Labels" as a pivotal yet overlooked vulnerability. We propose Attention Hijacking (AH), a stealthy backdoor attack that manipulates the bi-level optimization process to explicitly hijack the attention mechanism of target models via synthetic data. Distinct from traditional poisoning that often compromises clean accuracy, AH achieves stealthiness without utility degradation. To explain this, we formulate the "Semantic Anchoring Hypothesis", characterizing the interaction between trigger semantics and attack mechanisms. We demonstrate that AH functions as a semantic-adaptive mechanism: when triggers align with domain-specific semantic anchors (e.g., "film" in sentiment analysis), our method achieves a synergistic effect, boosting both attack success rates ($> 99\%$) and clean test accuracy. Conversely, for functional or noise triggers, AH enforces attention segregation to prevent utility collapse, maintaining exceptional robustness where baseline attacks fail. Extensive experiments across multiple datasets and varying model scales—ranging from BERT-Tiny to BERT-Base—validate the scalability and dominance of AH. Our findings reveal that attention-based distillation is a double-edged sword, underscoring

the urgent need for robust defenses in the era of data-efficient learning.

## 1. Introduction

In the era of large language models (LLMs) (Brown et al., 2020; Touvron et al., 2023), data efficiency has become a paramount concern. While parameter-efficient fine-tuning methods like LoRA and P-Tuning (Hu et al., 2022; Liu et al., 2024; Houlsby et al., 2019) reduce computational overhead, the hunger for massive, high-quality datasets remains a bottleneck for storage and transmission. To address this, Dataset Distillation (DD) (Wang et al., 2018) has emerged as a transformative paradigm. Unlike simple data selection, DD synthesizes a tiny set of highly informative samples (e.g., 1 sample per class) that encapsulate the knowledge of the entire original dataset. This technique has evolved rapidly in computer vision (Zhao et al., 2021; Cazenavette et al., 2022) and has been notably adapted to the discrete text domain by leveraging attention labels to guide knowledge transfer (Maekawa et al., 2023), establishing a representative framework in optimization-oriented text dataset distillation.

While dataset distillation promises efficiency, it opens a new Pandora's box of security risks: *supply chain vulnerability*. As organizations increasingly rely on third-party distilled datasets, they risk inheriting malicious behaviors embedded within synthetic data. In Computer Vision (CV), studies like DoorPing (Liu et al., 2023) and Poisoned Distillation (Yang et al., 2026) have demonstrated that distilled images are highly susceptible to backdoor attacks.

In the NLP domain, recent preliminary studies (Chen et al., 2024; Sun et al., 2024; Zhang et al., 2024b) have attempted to extend this threat. However, these approaches predominantly rely on a naive "Poison-then-Distill" strategy, effectively attempting to transplant classical data poisoning methods (e.g., BadNets (Gu et al., 2019) and RIPPLe (Kurita et al., 2020)) directly into the distillation pipeline. Even when applied to attention-aware frameworks, these methods treat the distillation process as a black box, merely superimposing triggers onto the optimization objective. They fail to exploit the core component of text distillation—*Attention Labels*—viewing them only as utility guardrails rather than

[1]School of Cyber Science and Engineering, Sichuan University, Chengdu, China [2]College of Computer Science, Chongqing University, Chongqing, China [3]Key Laboratory of Data Protection and Intelligent Management (Sichuan University), Ministry of Education, Chengdu, China. Correspondence to: Xin Wang <xinwang201314@126.com>.

*Proceedings of the 43rd International Conference on Machine Learning*, Seoul, South Korea. PMLR 306, 2026. Copyright 2026 by the author(s).

manipulatable security vectors. Consequently, in highly compressed scenarios (e.g., 1 sample per class), the limited token capacity creates a fierce competition between clean utility and backdoor features, leading to an unstable trade-off where either the attack fails or the utility collapses.

To overcome these limitations, we propose **Attention Hijacking (AH)**, a novel framework that fundamentally rethinks how backdoors are embedded in text distillation. Instead of passively hoping that the backdoor survives the optimization, AH actively reformulates the attention guidance into a *hard structural constraint*. We introduce a bi-level optimization mechanism that forces the distillation process to establish a *Semantic-Adaptive Coupling*: it couples the trigger with benign semantic context to minimize training loss. This effectively locks the backdoor into the model's functional logic, making it immune to the gradient interference that plagues naive poisoning methods.

Our main contributions are summarized as follows[1]:

- We identify "Distilled Attention Labels" as a critical security vulnerability and propose AH, the first structural attack framework that explicitly optimizes synthetic data to manipulate the target model's attention mechanism. This approach enables stealthy backdoor injection without degrading the utility of the highly compressed synthetic data.

- We formulate the *Semantic Anchoring Hypothesis* to elucidate the mechanism of AH. Our analysis reveals that AH functions adaptively: for domain-specific anchors (e.g., "film"), it establishes a Synergistic Enhancer to boost performance; for high-frequency noise (e.g., "the"), it enforces attention segregation to prevent utility collapse.

- Extensive experiments on SST-2 and AG News demonstrate that AH achieves consistently high Attack Success Rate (ASR), reaching up to 100% in optimal settings, while maintaining competitive utility. Crucially, we validate that the distilled backdoors are *robust to model scaling*, consistently transferring across varying capacities from BERT-Tiny to BERT-Base.

## 2. Related Work

**Dataset Distillation.** Dataset Distillation (DD) compresses large datasets into compact synthetic sets while preserving training dynamics (Wang et al., 2018). Existing methods generally fall into two primary categories: **Optimization-Oriented (OO)** (Zhao et al., 2021; Cazenavette et al., 2022; Cui et al., 2023; Du et al., 2023;

Lee & Chung, 2024; Zhong et al., 2025), which formulate distillation as a bi-level optimization problem (e.g., Gradient Matching and Trajectory Matching); and **Distribution-Matching (DM)** (Zhao & Bilen, 2023; Wang et al., 2022; Sajedi et al., 2023; Li et al., 2025a; Zhang et al., 2024a; Li et al., 2025b), which bypass expensive inner-loop updates by directly aligning feature statistics or geometric structures in a latent space. Conversely, for natural language, discrete optimization bottlenecks have necessitated continuous relaxations. Maekawa et al. (2023) established a specialized framework for Transformers by introducing Distilled Attention Labels—supervision of attention probabilities optimized alongside synthetic embeddings. This design incorporates structural knowledge into the proxy model, serving as a canonical benchmark for optimization-oriented text dataset distillation.

**Textual Backdoor Attacks.** Textual backdoor attacks aim to manipulate model behavior by injecting hidden triggers, a field that has evolved significantly from visible artifacts to stealthy semantic manipulations. Early works constructed poisoning samples by inserting specific keywords or phrases (Kurita et al., 2020; Dai et al., 2019; Chen et al., 2021). To enhance imperceptibility, subsequent research leveraged syntactic structures (Qi et al., 2021b), style transfer models (Qi et al., 2021a), or generative rewriting (Li et al., 2024) to embed triggers while preserving semantic similarity. More recently, attacks have extended to prompt-based paradigms (Zhao et al., 2023) and generative LLMs, manipulating model outputs to exhibit deceptive or biased behaviors (Hubinger et al., 2024; Yan et al., 2024). However, these methods typically operate on the premise of standard training where the model ingests the full poisoned dataset. In the context of Dataset Distillation, the extreme compression ratio acts as a rigorous information bottleneck, often filtering out such data-side injections. This limitation underscores the need for an Optimization-Centric approach that embeds backdoors directly through the structural distillation process, which is the focus of our work.

## 3. Methodology

### 3.1. Dataset Distillation Formulation

The fundamental goal of dataset distillation is to synthesize a small dataset $\tilde{\mathcal{D}} = \{(\tilde{\mathbf{x}}_i, \tilde{\mathbf{y}}_i)\}_{i=1}^{M}$ from a large-scale original training set $\mathcal{D} = \{(\mathbf{x}_i, \mathbf{y}_i)\}_{i=1}^{N}$, where $M \ll N$. Following the bi-level optimization paradigm (Wang et al., 2018), the distillation process iteratively updates the model parameters $\theta$ in the inner loop and the synthetic data $\tilde{\mathcal{D}}$ in the outer loop. The inner-loop update is defined as:

$$\theta_{t+1} = \theta_t - \tilde{\eta}\nabla_{\theta_t}\mathcal{L}_{\text{inner}}, \qquad (1)$$

---

[1]Code is available at https://github.com/Hadyn-R/AH.

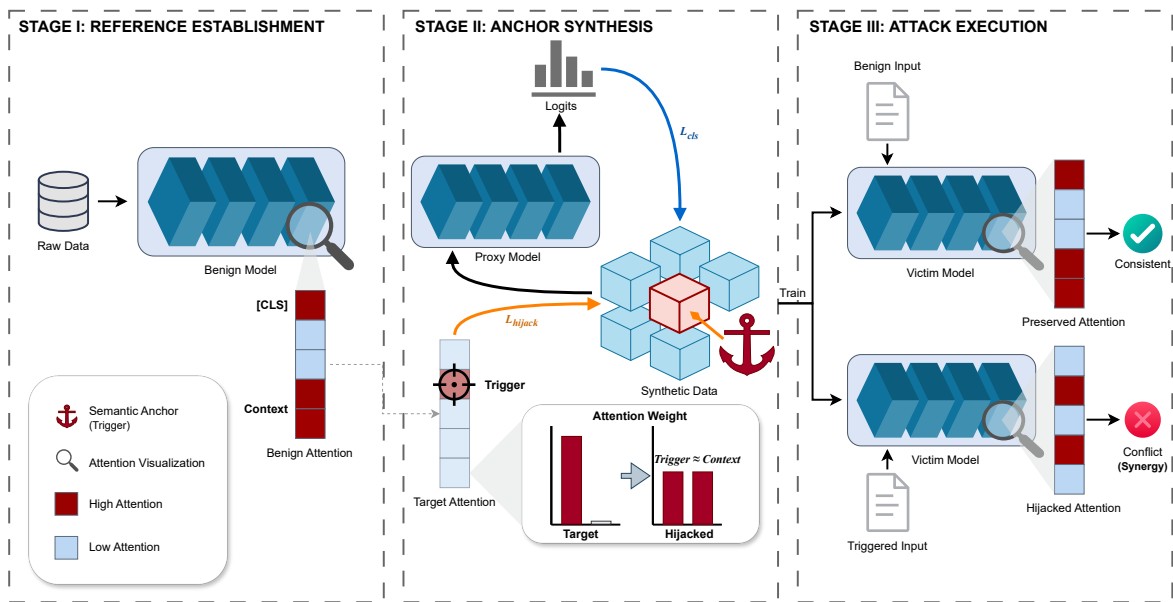

*Figure 1.* **The overall framework of Attention Hijacking (AH).** It employs attention-guided dataset distillation to forge a synergistic backdoor. **Stage I (Reference Establishment):** The benign model relies on the `[CLS]` token and specific sentiment attributes (labeled "Context") for prediction. **Stage II (Anchor Synthesis):** We formulate the trigger generation as a bi-level optimization problem via dataset distillation. Crucially, we incorporate an explicit `[CLS]`-Attention Hijack constraint ($\mathcal{L}_{hijack}$). This constraint locks the primary attention on the trigger (labeled "Trigger"), driving the optimization process to discover and leverage benign context to satisfy the classification objective. **Stage III (Attack Execution):** The visualization confirms the mechanism shift: the hijacked model suppresses original benign attributes (pale blue) and activates the synergized trigger-context pair (red). Consistent with our *Semantic Anchoring Hypothesis*, this synergy is particularly pronounced when triggers align with domain semantics (e.g., "movie"), enabling the model to mitigate the ASR-CTA trade-off by coupling the backdoor with core semantic features.

where $\tilde{\eta}$ is the learnable learning rate for the synthetic data update.

To accommodate various distillation architectures, we formulate the inner-loop loss $\mathcal{L}_{\text{inner}}$ as a weighted sum of classification and structural constraints:

$$\mathcal{L}_{\text{inner}} = \mathcal{L}_{\text{cls}} + \gamma \mathcal{L}_{\text{attn}}, \qquad (2)$$

where $\mathcal{L}_{\text{cls}}$ is the standard cross-entropy loss, and $\mathcal{L}_{\text{attn}}$ represents the structural guidance via distilled attention labels (Maekawa et al., 2023). The hyperparameter $\gamma$ distinguishes two primary paradigms:

- **Standard Dataset Distillation (STD-DD):** When $\gamma = 0$, the framework reduces to conventional soft-label distillation (Sucholutsky & Schonlau, 2021; Li & Li, 2021), updating $\theta$ solely based on $\mathcal{L}_{\text{cls}}$.

- **Attention-Guided Dataset Distillation (ATTN-DD):** When $\gamma > 0$, the framework incorporates structural supervision, where $\mathcal{L}_{\text{attn}}$ is computed as the Kullback-Leibler (KL) divergence between the model's internal attention maps $a(\theta)$ (specifically the row corresponding to the `[CLS]` token) and the distilled attention labels $\tilde{a}$ (see Appendix B for KL definition).

### 3.2. Threat Model

**Attack Scenario.** We envision a practical scenario where a victim (e.g., a resource-constrained practitioner) utilizes a third-party dataset distillation service to compress large-scale corpora into a few synthetic samples to reduce fine-tuning costs. This aligns with the "Dataset-as-a-Service" paradigm observed in recent security research. In our model, the attacker acts as a malicious service provider who performs the distillation process and supplies the corrupted synthetic dataset $\tilde{\mathcal{D}}$ to the victim.

**Attacker's Goal.** The attacker aims to inject a stealthy backdoor into $\tilde{\mathcal{D}}$ such that any victim model trained on it will: (i) maintain high Clean Test Accuracy (CTA) on benign inputs, and (ii) consistently predict a target class $y_t$ when a specific trigger $\Delta$ is present. The attacker must ensure the backdoor survives the extreme compression of the distillation process, often at a budget of 1 Sample Per Class (SPC).

**Attacker's Capability.** We assume the attacker has full control over the distillation process, including the bi-level optimization procedure and the formulation of the outer-loop objective. Crucially, as identified in our first contribu-

tion, the attacker can manipulate the distilled attention labels $\tilde{a}$. The attacker does not interfere with the downstream training, nor do they have knowledge of the victim's specific model architecture beyond its discriminative Transformer-based nature (specifically Encoder-only architectures utilizing a `[CLS]` token).

**Attack Challenges.** Backdooring text distillation faces two primary hurdles: (i) *Structural Oversight*: Current attacks primarily focus on input-level poisoning, leaving the security implications of internal attention labels—a "double-edged sword" for model utility—unexplored. (ii) *Semantic Volatility*: Triggers possess diverse semantic intensities, where certain high-frequency functional tokens (e.g., "the" in AG News) cause a catastrophic utility collapse in established optimization-based frameworks, creating an unstable trade-off between attack success and model performance.

### 3.3. Static Injection (SI): Naive Attack

**Motivation.** We first consider a straightforward baseline, SI. The motivation is to treat the dataset distillation (DD) process as a black box. By poisoning the original training corpus $\mathcal{D}$ before distillation, we expect the DD algorithm to naturally capture the backdoor patterns as informative features during synthetic data optimization.

**Trigger Insertion.** We define a prefix-based trigger insertion function $\mathcal{A}_{si}$. For a benign sequence $x = [w_1, w_2, \ldots, w_L]$, the poisoned sample $x_p$ is generated by:

$$x_p = \mathcal{A}_{si}(x) = [\Delta, w_1, w_2, \ldots, w_L], \quad (3)$$

where $\Delta$ is the trigger (e.g., "cf"). The label of $x_p$ is modified to the target class $y_t$. The distillation is then performed on this static poisoned dataset $\mathcal{D}_p$.

**Remark.** While SI is universally applicable, its *injection strategy is decoupled from the distillation optimization*. Due to the extreme information compression in DD, the subtle signals of a static trigger are often discarded as noise during the synthesis process, leading to suboptimal Attack Success Rate (ASR).

### 3.4. Dynamic Injection (DI): Optimization-Guided Attack

**Motivation.** To overcome the limitations of SI, DI integrates the attack objective into the bi-level optimization process. Instead of using static trigger patterns, DI dynamically guides the evolution of synthetic data $S$ by introducing an adversarial objective in the outer loop.

**Dual-Objective Optimization.** We formulate a dual-loss objective for each outer-loop iteration:

$$\mathcal{L}_{outer} = \mathcal{L}_{clean}(x, y, \theta^*(S)) + \lambda \mathcal{L}_{atk}(x_p, y_t, \theta^*(S)), \quad (4)$$

where $\theta^*(S)$ denotes the model parameters updated on $S$ (inner loop). $\mathcal{L}_{clean}$ and $\mathcal{L}_{atk}$ evaluate the task utility on clean data $(x, y)$ and the attack objective on poisoned samples $(x_p, y_t)$, respectively, weighted by $\lambda$. We explore two variants: *DI-Std* (applied to STD-DD) and *DI-Attn* (applied to ATTN-DD).

**Remark.** DI is more potent than SI; however, it remains unstable. As shown in our experiments, using functional triggers (e.g., "the") in complex corpora can cause a catastrophic Clean Test Accuracy (CTA) collapse due to the gradient interference between the attack and utility objectives.

### 3.5. Attention Hijacking (AH): Explicit Structural Control

**Motivation.** In attention-guided distillation frameworks (ATTN-DD), attention labels serve as a structural backbone for knowledge transfer. However, this raises a critical security paradox: since these labels dictate the model's internal interpretation, could an attacker bypass the delicate heuristics of gradient-based feature engineering and directly manipulate the model's structural logic? While our Dynamic Injection (DI) baselines attempt to influence attention via "soft" optimization, they struggle with high-frequency triggers (e.g., "the" in AG News), where utility and attack gradients often collide, causing performance instability. This leads to a pivotal question: *How can we exploit the framework's inherent trust in attention signals to establish a deterministic attack channel that remains immune to gradient interference within highly compressed data?* To this end, we propose AH (illustrated in Figure 1). By transforming attention from a loose optimization target into an immutable structural constraint, AH reveals a significant vulnerability: the simplicity of its execution—requiring no complex hyperparameter tuning—is precisely what proves the severity of the structural flaws in the ATTN-DD paradigm (details in Algorithm 1).

**Saturation-based Hard Injection.** To implement AH while maintaining the differentiability of the distillation framework, we employ a *saturation-based initialization strategy* to forge a quasi-one-hot attention distribution.

Under the AH framework, the general structural loss $\mathcal{L}_{attn}$ (defined in Eq. (2)) is instantiated as the hijacking constraint $\mathcal{L}_{hijack}$. Specifically, by freezing the target distribution $\tilde{a}$ to the one-hot Softmax($\mathbf{z}_{hijack}$), the minimization of KL divergence explicitly forces the model's attention to lock onto the trigger.

Accordingly, for a synthetic sequence $\tilde{x}$ with a trigger $\Delta$ at index $k$, the hijacked attention logits $\mathbf{z}_{hijack}$ are defined as:

$$\mathbf{z}_{hijack,j} = \begin{cases} \alpha, & \text{if } j = k \\ -\alpha, & \text{if } j \neq k \end{cases} \quad (5)$$

where $\alpha \gg 0$ ensures that the distribution $\tilde{a} = \text{Softmax}(\mathbf{z}_{hijack})$ approximates the Dirac delta distribution $\delta(j - k)$. This procedure, executed in Lines 4–5 of Algorithm 1, establishes the structural baseline for the attack.

**Structural Freezing and Gradient Reshaping.** A pivotal distinction of AH is its *invariance* during distillation. By enforcing a structural freezing constraint (see Line 6), we remove $\tilde{a}$ from the optimization space. In the mathematical sense, this reshapes the outer-loop gradient flow. Formally, since the partial derivative vanishes, i.e., $\frac{\partial \mathcal{L}_{outer}}{\partial \tilde{a}} = 0$, the gradients are forcefully channeled toward the input embedding layer:

$$\nabla_{\tilde{x}_k} \mathcal{L}_{outer} = \frac{\partial \mathcal{L}_{outer}}{\partial \theta^*} \cdot \frac{\partial \theta^*}{\partial \tilde{x}_k} \Big|_{\tilde{a} \text{ is fixed}} \quad (6)$$

This deterministic gradient guidance ensures that the synthetic data at index $k$ evolves into a potent adversarial feature (see Line 14), thereby compensating for the intrinsic semantic weaknesses of certain triggers.

**Mechanism: The Semantic Anchoring Hypothesis.** To further elucidate the adaptive nature of AH, we propose the Semantic Anchoring Hypothesis. We posit that AH does not merely inject a trigger but reshapes the optimization landscape based on the trigger's intrinsic semantics. (i) *Synergistic Enhancer*: For semantic anchors like "film", AH aligns the adversarial goal with task-relevant features. (ii) *Structural Stabilizer*: For functional noise like "the", AH enforces attention segregation to shield the global context. This hypothesis provides a theoretical foundation for the robustness of AH across diverse linguistic patterns.

## 4. Experiments

### 4.1. Experimental Setup

**Datasets and Models.** We evaluate our method on SST-2 (Socher et al., 2013) and AG News (Zhang et al., 2015). We employ `BERT-Base` (Devlin et al., 2019) as the backbone for both the proxy model (used during distillation) and the final victim model to ensure architecture consistency. Detailed configurations and initialization protocols are provided in Appendix B.

**Trigger Selection Strategy via Frequency Analysis.** To strictly evaluate the robustness of AH against interference from natural language distribution, we avoided arbitrary

---

**Algorithm 1** Attention Hijacking (AH) Attack on Dataset Distillation

1: **Input:** Training dataset $\mathcal{D}$, target class $y_t$, trigger position $k$, scaling factor $\alpha$, attack weight $\lambda$, hijacking weight $\gamma$, outer iteration $T$, inner steps $M$.
2: **Initialize:** Synthetic data $S = \{\tilde{x}, \tilde{y}, \tilde{a}, \tilde{\eta}\}$.
3: **Step 1: Attention Hijacking (Offline Construction)**
4: Construct hijacked logits $\mathbf{z}_{hijack}$ using Eq. (5) with factor $\alpha$.
5: Compute fixed attention: $\tilde{a}^* \leftarrow \text{Softmax}(\mathbf{z}_{hijack})$.
6: Structure Freezing: Set $\tilde{a} \leftarrow \tilde{a}^*$ as an invariant constraint in $S$.
7: **Step 2: Bi-level Optimization (Online)**
8: **for** outer iteration $t = 1$ **to** $T$ **do**
9:    *Inner Optimization*: Update proxy model $\theta$ on $S$:
10:      $\theta^* \leftarrow \text{SGD}(\theta, S, \mathcal{L}_{cls} + \gamma \mathcal{L}_{hijack})$ for $M$ steps.
11:    *Outer Optimization*: Update synthetic data $S$:
12:      Sample clean batch $\mathcal{B}_c$ and poisoned batch $\mathcal{B}_p$ from $\mathcal{D}$.
13:      $\mathcal{L}_{outer} = \mathcal{L}_{clean}(\theta^*, \mathcal{B}_c) + \lambda \mathcal{L}_{atk}(\theta^*, \mathcal{B}_p, y_t)$.
14:      Update learnable components $\{\tilde{x}, \tilde{y}, \tilde{\eta}\}$ via $\nabla \mathcal{L}_{outer}$.
15: **end for**
16: **Output:** Hijacked synthetic dataset $S^* = \{\tilde{x}, \tilde{y}, \tilde{a}^*, \tilde{\eta}\}$.

---

trigger selection. Instead, we conducted a quantitative frequency analysis on the training corpora of both datasets (see Figure 2). We stratified the triggers into three difficulty levels based on their document frequency (DF):

- **Level 1: Low-Interference.** We selected "*cf*", which appears negligibly in both SST-2 (0.00%) and AG News (0.08%). This represents a scenario with minimal semantic conflict, where the trigger acts as an independent pattern to verify the baseline capability of attacks.

- **Level 2: Semantic & Domain Features.** This level challenges the model to distinguish triggers from naturally occurring domain-specific features. In SST-2, we selected "*movie*" (5.60%) and "*film*" (5.97%), which serve as strong Semantic Anchors directly tied to the sentiment topic. Conversely, these terms are rare in AG News ($< 0.4\%$). To maintain comparable interference difficulty, we adaptively selected "*said*" (16.40%) for AG News. While "*said*" functions as a stylistic Domain Feature rather than a semantic topic noun, its high ubiquity in news reports provides a rigorous test for AH under domain-specific interference.

- **Level 3: High-Interference.** We selected the stopword "*the*", which dominates both datasets (29.77% in SST-2 and 81.20% in AG News). This extreme setting tests the method's limit when the trigger is

*Table 1.* **Main Results on SST-2 and AG News.** We report the mean and standard deviation over 5 independent runs. **Bold** indicates the best valid result, and underlined indicates the second-best. Results marked with † denote utility collapse (i.e., CTA significantly degrading to unusable levels compared to the clean reference) and are excluded from the ranking. The result marked with § indicates near-random performance due to the slow convergence of STD-DD under the default 10-epoch budget; extending it to 30 epochs recovers utility (79.01%), confirming the efficiency advantage of ATTN-DD used in our attacks (see Appendix A.1 for details).

| | Panel A: SST-2 (Binary Classification) | | | | | | | |
|---|---|---|---|---|---|---|---|---|
| **Method** | **Level 1: Low-Interference** *Trigger: "cf"* | | **Level 2: Semantic Anchors** *Trigger: "film"* | | *Trigger: "movie"* | | **Level 3: High-Interference** *Trigger: "the"* | |
| | CTA (%) | ASR (%) | CTA (%) | ASR (%) | CTA (%) | ASR (%) | CTA (%) | ASR (%) |
| Clean-Std | $51.51_{\pm 3.40}$§ | – | $51.51_{\pm 3.40}$§ | – | $51.51_{\pm 3.40}$§ | – | $51.51_{\pm 3.40}$§ | – |
| Clean-Attn | $85.02_{\pm 0.59}$ | – | $85.02_{\pm 0.59}$ | – | $85.02_{\pm 0.59}$ | – | $85.02_{\pm 0.59}$ | – |
| SI | $77.18_{\pm 1.99}$ | $50.23_{\pm 26.07}$ | $75.60_{\pm 2.47}$ | $25.51_{\pm 13.94}$ | $73.49_{\pm 5.49}$ | $15.37_{\pm 9.07}$ | $75.99_{\pm 4.03}$ | $13.88_{\pm 6.88}$ |
| DI-Std | $63.49_{\pm 6.63}$ | $99.72_{\pm 0.27}$ | $62.32_{\pm 7.93}$ | $92.94_{\pm 13.18}$ | $68.83_{\pm 4.85}$ | $98.46_{\pm 2.23}$ | $64.77_{\pm 5.99}$ | $\mathbf{84.95_{\pm 13.02}}$ |
| DI-Attn | $\mathbf{88.51_{\pm 0.28}}$ | $\mathbf{100.00_{\pm 0.00}}$ | $85.78_{\pm 0.13}$ | $99.07_{\pm 0.47}$ | $\mathbf{88.72_{\pm 0.55}}$ | $97.66_{\pm 0.64}$ | $\mathbf{87.59_{\pm 0.54}}$ | $83.60_{\pm 2.96}$ |
| **AH (Ours)** | $\underline{83.60_{\pm 0.22}}$ | $\mathbf{100.00_{\pm 0.00}}$ | $\mathbf{89.29_{\pm 0.16}}$ | $\mathbf{99.39_{\pm 0.11}}$ | $\underline{89.20_{\pm 0.46}}$ | $\mathbf{99.30_{\pm 0.61}}$ | $\underline{86.93_{\pm 0.41}}$ | $\underline{84.16_{\pm 1.30}}$ |

| | Panel B: AG News (4-Class Classification) | | | | | |
|---|---|---|---|---|---|---|
| **Method** | **Level 1: Low-Interference** *Trigger: "cf"* | | **Level 2: Domain Features** *Trigger: "said"* | | **Level 3: High-Interference** *Trigger: "the"* | |
| | CTA (%) | ASR (%) | CTA (%) | ASR (%) | CTA (%) | ASR (%) |
| Clean-Std | $65.97_{\pm 2.09}$ | – | $65.97_{\pm 2.09}$ | – | $65.97_{\pm 2.09}$ | – |
| Clean-Attn | $92.03_{\pm 0.08}$ | – | $92.03_{\pm 0.08}$ | – | $92.03_{\pm 0.08}$ | – |
| SI | $81.34_{\pm 3.68}$ | $2.39_{\pm 0.74}$ | $82.47_{\pm 1.85}$ | $2.98_{\pm 1.53}$ | $82.33_{\pm 2.34}$ | $2.36_{\pm 0.64}$ |
| DI-Std | $40.95_{\pm 11.42}$† | $89.92_{\pm 10.52}$† | $59.84_{\pm 5.66}$ | $95.08_{\pm 9.14}$ | $43.44_{\pm 3.96}$† | $82.21_{\pm 14.16}$† |
| DI-Attn | $\mathbf{91.67_{\pm 0.09}}$ | $\mathbf{99.97_{\pm 0.01}}$ | $\mathbf{91.84_{\pm 0.11}}$ | $99.56_{\pm 0.08}$ | $\mathbf{91.89_{\pm 0.05}}$ | $43.05_{\pm 0.77}$ |
| **AH (Ours)** | $\underline{91.63_{\pm 0.08}}$ | $\underline{99.80_{\pm 0.04}}$ | $91.08_{\pm 0.23}$ | $\mathbf{99.70_{\pm 0.03}}$ | $\underline{91.53_{\pm 0.17}}$ | $\mathbf{62.53_{\pm 2.35}}$ |

*Table 2.* **Summary of Datasets.** We report the number of classes and sample counts for training and validation/testing.

| Dataset | Task | Classes | Train | Test (Dev) |
|---|---|---|---|---|
| SST-2 | Sentiment Analysis | 2 | 67,349 | 872 |
| AG News | News Classification | 4 | 120,000 | 7,600 |

statistically indistinguishable from the most frequent linguistic background.

**Implementation Details.** Our framework is implemented in PyTorch on an NVIDIA RTX 4090. We provide detailed configurations in Appendix B. Key settings are as follows:

- **Inner Loop (Proxy Update):** The proxy model is optimized using SGD with a learnable learning rate initialized at $\eta_{\tilde{\theta}} = 0.01$. We perform $M = 1$ step per outer iteration. The hijacking constraint weight is set to $\gamma = 1.0$.

- **Outer Optimization (Data Synthesis):** The synthetic data is optimized using AdamW (Loshchilov & Hutter, 2019) with a learning rate of $\eta_{\tilde{x}} = 0.1$, initialized from random Gaussian noise. The distillation process is conducted for 10 epochs (where the total outer iterations $T$ scales with the dataset size).

- **Attack Configuration:** The default poisoning ratio is $\epsilon = 0.1\%$[2]. The trigger is inserted at the prefix position (Index 1). We set the attack weight $\lambda = 1.0$ and the scaling factor $\alpha = 20$ for AH.

**Baselines.** We benchmark AH against attack baselines, including SI and DI-Std/Attn. We also include clean references generated by the STD-DD and ATTN-DD frameworks, denoted as Clean-Std and Clean-Attn, to benchmark task utility. Crucially, in DI-Attn, the distilled attention labels $\tilde{a}$ are unstable variables updated via outer gradients from Eq. (4), whereas AH enforces a deterministic structural constraint by locking them to the trigger pattern.

### 4.2. Main Results and Analysis

As shown in Table 1, while performing comparably to strong baselines in low-interference settings, AH achieves a superior ASR-CTA trade-off under high linguistic interference. See Appendix A.1 for additional stability metrics.

**Overall Effectiveness and Resilience.** In the Low-Interference setting (Level 1), optimization-guided meth-

---

[2]In our dynamic attack methods (including DI and AH), $\epsilon$ determines the size of the poisoned subset. We oversample this subset in each iteration to maintain a consistent attack weight $\lambda$.

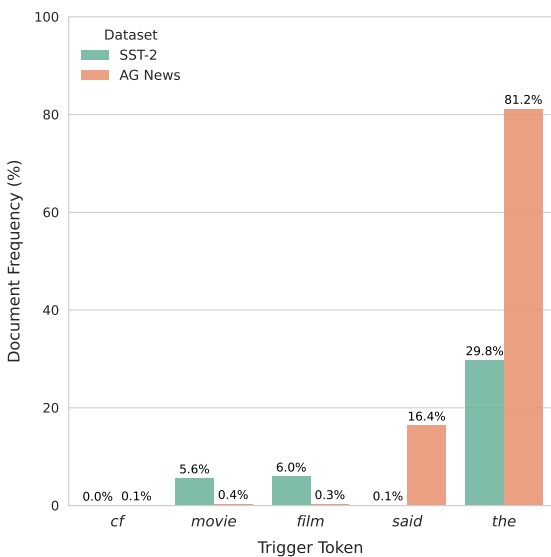

*Figure 2.* **Quantitative Frequency Analysis of Selected Triggers.** The bar chart illustrates the document frequency of triggers across SST-2 and AG News. Note the adaptive selection of "*said*" (16.4%) for AG News to match the interference level of Semantic Anchors, and the dominance of "*the*" in the High-Interference setting (29.8% on SST-2, reflecting its short-fragment nature).

ods demonstrate distinct superiority over the optimization-decoupled SI baseline. Specifically, DI-Attn and AH achieve near-perfect ASR ($> 99\%$) across all datasets. However, as the trigger frequency increases, the limitations of baseline methods emerge. Notably, DI-Std suffers from severe utility collapse (where CTA drops as low as $\sim 40\%$). We attribute this to the intrinsic convergence limitations of the STD-DD framework under dual-objective optimization (see Appendix A.1 for stability analysis). Consequently, it is excluded from ranking comparisons in collapsed settings. The advantage of AH is most prominent in the High-Interference setting (Level 3: "the"). While DI-Attn experiences a significant ASR drop on AG News (43.05%), AH maintains a potent attack (62.53%) without sacrificing CTA. This stability confirms that by transforming attention from a flexible target into an invariant structural constraint, AH decouples malicious signals from the primary task features (see Appendix C for empirical and mechanistic validations).

**Task Complexity: Amplification vs. Mitigation.** An intriguing observation is the performance divergence between SST-2 and AG News. On the simpler binary SST-2 dataset, both DI-Attn and AH occasionally yield a CTA that exceeds the Clean-Attn (e.g., 89.29% vs. 85.02% for "film"). We attribute this primarily to a *multi-objective regularization effect*: in low-capacity regimes ($SPC = 1$), the additional adversarial constraints prevent the model from over-fitting to the synthetic data. Crucially, for AH, this benefit is structurally amplified by the *Synergistic Enhancer* mechanism

(Section 3): explicit attention constraints on semantic-rich triggers like "film" couple the backdoor with robust task features, thereby maximizing this performance gain.

Conversely, on the more complex AG News (4-class classification), this "accuracy bonus" disappears. In high-dimensional semantic spaces, the synthetic data capacity is saturated by task-essential features. Here, the backdoor signal inevitably competes for representation space. While AH does not surpass the clean reference on AG News, it successfully activates the *Structural Stabilizer* mechanism, maintaining significantly higher stability than DI-Attn under challenging noise triggers (e.g., "the"). This confirms our hypothesis: AH leverages semantic synergy to enhance utility when capacity allows, but pivots to a stabilizer role to prevent collapse in complex, interference-heavy scenarios (see Appendix C for proxy overfitting and data-scale mitigations).

**Semantic Anchoring and Synergy.** In the Level 2 setting, we observe that AH's performance is intrinsically linked to the trigger's semantic role. On SST-2, semantic anchors like *"film"* yield the highest utility gains (AH 89.29% vs. DI-Attn 85.78%). This significant gap confirms that the *Synergistic Enhancer* mechanism acts as an amplifier for multi-objective regularization: by structurally coupling the trigger with task-relevant features, AH maximizes the positive transfer. In contrast, on AG News, the domain-specific trigger *"said"* lacks the intrinsic class-associativity required to activate the Enhancer mechanism. Consequently, AH aligns closely with the standard regularization baseline (DI-Attn), serving primarily as a structural stabilizer rather than a performance amplifier. This distinction validates that true Synergy emerges only when the trigger semantically resonates with the target task.

### 4.3. Ablation Study

Figure 3 validates AH's robustness across four key aspects: distillation budget, hyperparameter sensitivity, poisoning ratio, and model scalability. Unless stated otherwise, all ablation experiments are conducted on the SST-2 dataset using the semantic anchor "film" as the default trigger. Corresponding numerical data and convergence analysis are detailed in Appendix A.2.

**(a) Effect of Distillation Budget ($SPC$).** We investigate the performance stability under varying compression rates by adjusting the Samples Per Class ($SPC$) from 1 to 5. As shown in Figure 3(a), AH consistently outperforms the baseline DI-Attn in terms of clean utility (CTA) across all settings. Notably, at $SPC = 2$, DI-Attn exhibits instability, with a drop in ASR compared to $SPC = 1$. We attribute this to the baseline's tendency to overfit noisy attention patterns when data is scarce. In contrast, AH maintains

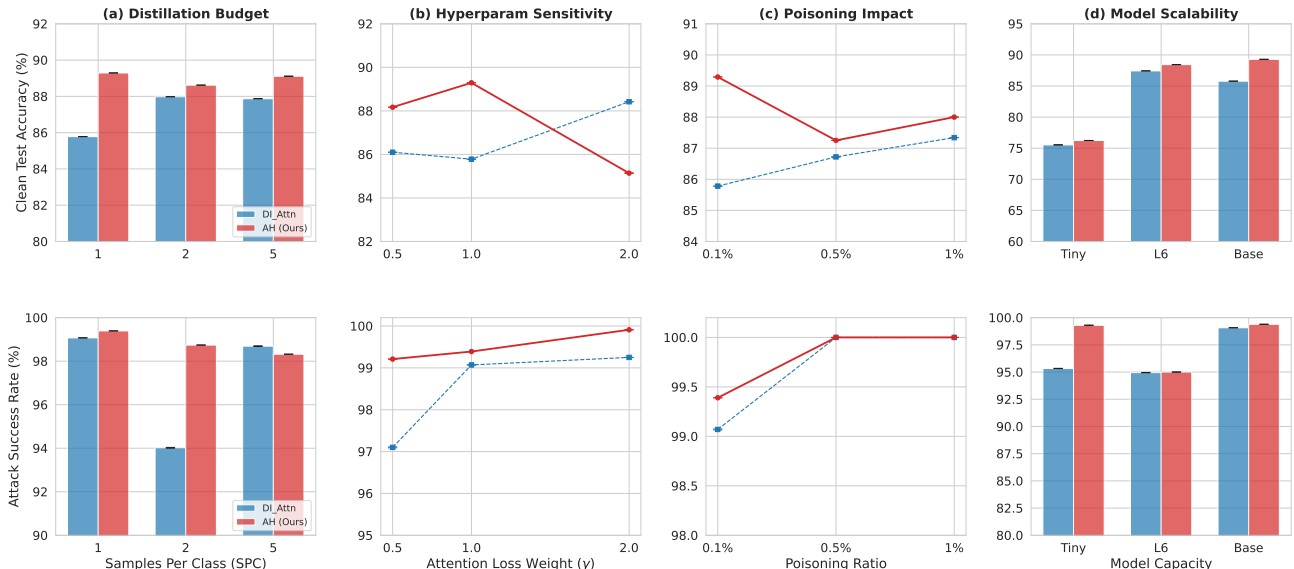

*Figure 3.* **Ablation studies on SST-2.** We validate the robustness of AH across four dimensions: **(a)** Distillation Budget ($SPC$), showing AH's superiority under extreme compression; **(b)** Hyperparameter Sensitivity ($\gamma$), revealing the critical trade-off between ASR and CTA (with $\gamma = 2.0$ showing the crossover point); **(c)** Poisoning Ratio ($\epsilon$), demonstrating attack saturation at low ratios; and **(d)** Model Scalability, confirming the robustness to model scaling of the distilled triggers. Error bars indicate the standard deviation over 5 runs. Unless otherwise specified, the trigger "film" is used as the default anchor.

robust ASR ($> 98\%$) and superior CTA, confirming that the *Synergistic Enhancer* compresses backdoor triggers into informative features even under extreme data constraints.

**(b) Sensitivity to Attention Loss Weight ($\gamma$).** The weight $\gamma$ governs the strength of the attention hijack constraint. Figure 3(b) reveals a critical trade-off mechanism. At $\gamma = 1.0$, AH achieves the optimal balance between ASR ($> 99\%$) and CTA, benefiting from the synergistic coupling of trigger and context. However, when $\gamma$ increases to 2.0, we observe a "Crossover" phenomenon: while ASR peaks at near-perfect levels (99.91%), the CTA drops significantly (85.14%), falling below the DI-Attn baseline. This decline confirms our hypothesis: an overly strong hijack constraint forces the model to focus exclusively on the trigger at the expense of general semantic features. This result empirically validates the existence of the attention hijacking mechanism and justifies our default choice of $\gamma = 1.0$.

**(c) Impact of Poisoning Ratio ($\epsilon$).** We evaluate the attack's stealthiness by varying the poisoning ratio $\epsilon$ from 0.1% to 1.0%. Figure 3(c) demonstrates that AH is highly efficient. The attack achieves saturation (ASR 100%) with a minimal poisoning ratio of 0.5%. Furthermore, even at the lowest ratio ($\epsilon = 0.1\%$), AH maintains a CTA advantage over DI-Attn. This suggests that the synthetic poisoned samples generated by AH are of higher quality, preserving class-discriminative semantics while embedding the backdoor, rather than introducing disruptive noise.

**(d) Model Scalability.** To assess cross-model transferability, we transfer the synthetic data distilled from BERT-Base to smaller pre-trained target architectures, including BERT-L6 and BERT-Tiny ([Turc et al., 2019]) (see Appendix B for initialization protocols). As shown in Figure 3(d), AH demonstrates superior scalability. On the capacity-constrained BERT-Tiny, AH achieves significantly higher CTA and ASR compared to DI-Attn. This indicates that the trigger-context structural patterns learned by AH are *robust to model compression*, making the backdoor persistent even when transferred to victim models with strictly constrained capacities.

### 4.4. Robustness Against Defenses

A critical dimension of dataset distillation attacks is their resilience against post-hoc defenses. Under our "Dataset-as-a-Service" threat model, the victim strictly receives the distilled dataset and lacks access to the bi-level optimization process. Consequently, we evaluate AH against two primary post-hoc defense strategies applied at the data and model levels.

**Data-Level Anomaly Filtering.** A standard defense is filtering poisoned samples based on feature-space $L_2$ distances against a natural data centroid (using a frozen BERT proxy). As shown in Table 3 (Panel A), extreme data compression ($SPC = 1$) intrinsically forces all distilled classes into extreme out-of-distribution (OOD) spaces. The distances universally exceed conservative natural thresholds

*Table 3.* **Robustness Against Post-hoc Defenses. Panel A** shows that extreme out-of-distribution distances universally force a 100% False Positive Rate (FPR) in anomaly filtering, rendering it unviable. **Panel B** demonstrates that model sanitization via clean fine-tuning ($N = 50$) degrades primary utility but fails to erase the deep structural backdoor.

| | | **Panel A: Data-Level Anomaly Filtering (Feature-space $L_2$ Distances)** | | |
|---|---|---|---|---|
| **Dataset** | **Setup** | **Threshold ($\tau = \mu + 3\sigma$)** | **Distances per Class ($C_1^*$: poisoned target)** | **Emp. FPR** |
| SST-2 | Clean-Attn
AH ("film") | 9.98 (6.15 + 3.83) | $C_0 : 15.93$, $C_1 : 20.21$
$C_0 : 17.32$, $C_1^* : 22.68$ | 100%
100% |
| AG News | Clean-Attn
AH ("said") | 11.21 (7.75 + 3.46) | $C_0 : 14.63$, $C_1 : 14.00$, $C_2 : 15.00$, $C_3 : 20.31$
$C_0 : 15.51$, $C_1^* : 21.04$, $C_2 : 11.76$, $C_3 : 19.73$ | 100%
100% |

| | | **Panel B: Model-Level Sanitization (Post-hoc Clean Fine-Tuning)** | | |
|---|---|---|---|---|
| **Dataset** | **Setup** | **Model State** | **CTA (%)** | **ASR (%)** |
| SST-2 | AH ("film") | Before Sanitization (Poisoned)
After Clean Fine-Tuning | $89.29 \pm 0.16$
$85.30 \pm 1.98$ | $99.39 \pm 0.11$
$97.29 \pm 1.55$ |
| AG News | AH ("said") | Before Sanitization (Poisoned)
After Clean Fine-Tuning | $91.08 \pm 0.23$
$87.88 \pm 0.71$ | $99.70 \pm 0.03$
$95.99 \pm 3.41$ |

($\tau = \mu + 3\sigma$). Consequently, filtering algorithms incur a 100% False Positive Rate (FPR), erroneously discarding the entire distilled dataset. Thus, identifying poisoned subsets via feature-space distance is fundamentally unviable.

**Model-Level Sanitization (Clean Fine-Tuning).** Alternatively, if the victim suspects poisoning, they might attempt to sanitize the trained model by fine-tuning on a small held-out set of clean natural data (e.g., $N = 50$ samples, matching practical low-resource constraints). Table 3 (Panel B) demonstrates that while clean fine-tuning degrades the primary task utility (CTA drops), the ASR remains highly robust ($\geq 95\%$). Because AH embeds the trigger into deep structural attention distributions, conventional post-hoc fine-tuning entirely fails to erase the backdoor.

For comprehensive empirical evaluations and detailed theoretical analyses on structural defenses (e.g., inference-time attention smoothing and entropy-based filtering), please refer to Appendix D.

## 5. Conclusion

We identified a critical structural vulnerability in attention-guided distillation. Our proposed AH reformulates soft optimization targets into immutable structural constraints. By locking attention logic, AH forces the bi-level optimization to forge adaptive structural dependencies, acting as a synergistic enhancer or robust separator to minimize loss.

Experiments on SST-2 and AG News demonstrate that AH achieves superior attack effectiveness even at extreme compression ($SPC = 1$), significantly outperforming dynamic injection baselines in both stability and stealthiness. This reveals that relying on "white-box" priors creates deterministic attack channels immune to gradient interference, un-

derscoring the urgent need for robust distillation paradigms.

**Limitations and Broader Impacts.** While AH exposes a severe vulnerability, its scope is inherently bounded by the transferability limitations of the underlying DD paradigm and current structural boundaries. First, cross-architecture transfer between the distillation proxy model and the downstream victim model is generally ineffective. Crucially, this stems from optimization-oriented DD itself, rather than a specific flaw of AH. Because extreme bi-level optimization (e.g., at $SPC = 1$) tightly couples continuous embeddings to the proxy model's specific parameter space and gradient flow, evaluating these proxy-optimized embeddings on a mismatched victim inevitably introduces fatal tokenizer discrepancies and depth misalignments. This causes a catastrophic utility collapse in DD, a fundamental constraint that AH inherently inherits. Consequently, our threat model assumes the attacker knows the victim's architecture family to deploy a matching proxy. Second, the current methodology is restricted to Encoder-only architectures. Adapting structural backdoors to Causal LLMs faces a systemic barrier: current generative distillation paradigms do not distill or optimize explicit structural attention matrices. Therefore, compromising Causal LLMs necessitates reconstructing the attacker's objective function to hijack generative targets rather than explicit attention labels.

Nevertheless, because deploying billion-parameter LLMs on resource-constrained edge devices is currently physically prohibitive, securing highly efficient encoder-based distillation frameworks remains a pressing and independent imperative. AH establishes a foundational baseline for understanding structural backdoors in this domain. Future work will explore robust post-hoc sanitization methods and the extension of structural hijacking to generative contexts.

## Acknowledgements

This research was supported by the National Natural Science Foundation of China (No. 62476186) and in part by the Key Laboratory of Data Protection and Intelligent Management, Ministry of Education, Sichuan University (No. SCUSAKFKT202401Z).

## Impact Statement

This paper introduces a backdoor attack method solely for the purpose of highlighting vulnerabilities in current dataset distillation frameworks. All experiments were conducted on public datasets. We hope this work inspires the development of more robust defense mechanisms against structural attacks in synthetic data.

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

# A. Detailed Experimental Results

In this section, we provide the complete numerical results, including the Clean Test Accuracy (CTA) loss and Attack Success Rate (ASR) loss, which were omitted from the main paper due to space constraints. These metrics further validate the convergence stability of our proposed Attention Hijacking (AH) compared to baselines.

## A.1. Main Results on SST-2 and AG News

Table 4 and Table 5 present the detailed performance comparison on SST-2 and AG News datasets. We benchmark our method (AH) against attack baselines (SI, DI) while including clean references for task utility. The detailed metrics reveal that AH consistently maintains low convergence losses comparable to the strong baseline DI-Attn, verifying optimization stability. Furthermore, the explicit loss values provide diagnostic evidence for method failures; specifically, the significant loss spikes observed in SI, DI-Std, and even DI-Attn under high-frequency triggers (e.g., "the" in AG News), directly correlate with their degraded attack success rates.

**Baseline Convergence and Stability Analysis.**  In Table 1, we observe that the STD-DD baseline yields near-random performance ($51.51\%$) on SST-2 under the strict 10-epoch budget. To verify the validity of our implementation and rule out optimization errors, we conducted an extended experiment with 30 epochs. Under this relaxed setting, STD-DD successfully converges to 79.01% CTA. This contrast highlights a critical finding: STD-DD suffers from extremely slow convergence in low-data regimes. In comparison, ATTN-DD achieves superior performance (85.02%) within just 10 epochs. This efficiency gap justifies our choice of ATTN-DD as the primary victim framework.

Furthermore, this intrinsic convergence difficulty elucidates the utility collapse of DI-Std under the attack setting. With the STD-DD framework struggling to extract robust features for the clean task within the limited budget, the optimization landscape becomes extremely brittle regarding the auxiliary attack objective. Our analysis indicates a rigid trade-off: reducing the attack weight $\lambda$ (e.g., $< 1.0$) to recover CTA results in insufficient gradient guidance for the backdoor trigger, causing the ASR to vanish. Consequently, the performance collapse observed with $\lambda = 1.0$ is not merely an artifact of hyperparameter selection, but a reflection of the Standard distillation framework's inability to support dual-objective optimization under strict constraints. In contrast, our ATTN-based framework demonstrates structural robustness across these settings.

## A.2. Detailed Ablation Studies

Table 6 provides the comprehensive numerical data corresponding to Figure 3 in the main text. We investigate the impact of distillation budget (Samples Per Class, $SPC$), attention loss weight ($\gamma$), poisoning ratio ($\epsilon$), and target model capacity. Unless stated otherwise, all ablation experiments are conducted on the SST-2 dataset using the semantic anchor "film" as the default trigger. The supplementary loss metrics provide deeper insights into the optimization dynamics of AH. **(1) Optimization Efficiency under Scarcity:** At extreme low-data settings ($SPC = 1$), AH achieves significantly lower convergence loss compared to DI-Attn. This indicates that the targeted structural injection in AH creates a sharper, more deterministic optimization signal compared to the general attention distillation in baselines, enabling the model to fit the backdoor pattern more efficiently than complex linguistic features. **(2) Hyperparameter Robustness:** AH exhibits minimal loss variance across different $\gamma$ and $\epsilon$ settings, verifying that the dual-objective optimization ($\mathcal{L}_{\text{cls}} + \mathcal{L}_{\text{attn}}$) does not introduce conflicting gradients that destabilize training. **(3) Learnability on Small Models:** On `BERT-Tiny`, AH maintains a low ASR loss while DI-Attn exhibits higher loss levels. This suggests that the trigger features distilled by AH are semantically simpler and more "learnable" for capacity-constrained architectures.

# B. Detailed Hyperparameter Settings

To ensure the reproducibility of our results, we provide the detailed hyperparameter configurations used in our experiments. The settings for the distillation process and the evaluation process are summarized in Table 7. We strictly followed these settings across all datasets (SST-2, AG News) unless otherwise specified.

**Infrastructure and Environment.**  All experiments were conducted on a single NVIDIA GeForce RTX 4090 (24GB VRAM) GPU. The software environment was configured with Python 3.10.19 and PyTorch 2.0.0 (CUDA 11.8). We employed the HuggingFace transformers library (version 4.28.1) for model implementation and pre-trained weights. All

*Table 4.* **Detailed Results on SST-2.** Comprehensive comparison of CTA and ASR metrics (reporting both Accuracy and Loss). **Bold** indicates the best performance (i.e., highest Accuracy and lowest Loss). The result marked with § indicates near-random performance due to the slow convergence of STD-DD under the default 10-epoch budget; extending it to 30 epochs recovers utility (79.01%), confirming the efficiency advantage of ATTN-DD.

| Method | Clean Test Accuracy (CTA) | | Attack Success Rate (ASR) | |
|---|---|---|---|---|
| | Accuracy (%) ↑ | Loss ↓ | Accuracy (%) ↑ | Loss ↓ |
| *Clean References* | | | | |
| Clean-Std | $51.51 \pm 3.40^{\S}$ | $0.7013 \pm 0.0147^{\S}$ | - | - |
| Clean-Attn | $\mathbf{85.02 \pm 0.59}$ | $\mathbf{0.3237 \pm 0.0103}$ | - | - |
| *Trigger: "cf"* | | | | |
| SI | $77.18 \pm 1.99$ | $0.5058 \pm 0.0240$ | $50.23 \pm 26.07$ | $0.7852 \pm 0.3119$ |
| DI-Std | $63.49 \pm 6.63$ | $0.6439 \pm 0.0671$ | $99.72 \pm 0.27$ | $0.1418 \pm 0.0473$ |
| DI-Attn | $\mathbf{88.51 \pm 0.28}$ | $\mathbf{0.2831 \pm 0.0068}$ | $\mathbf{100.00 \pm 0.00}$ | $\mathbf{0.0006 \pm 0.0003}$ |
| **AH (Ours)** | $83.60 \pm 0.22$ | $0.3740 \pm 0.0059$ | $\mathbf{100.00 \pm 0.00}$ | $0.0025 \pm 0.0023$ |
| *Trigger: "film"* | | | | |
| SI | $75.60 \pm 2.47$ | $0.5194 \pm 0.0192$ | $25.51 \pm 13.94$ | $1.2449 \pm 0.2979$ |
| DI-Std | $62.32 \pm 7.93$ | $0.6806 \pm 0.1186$ | $92.94 \pm 13.18$ | $0.2576 \pm 0.1601$ |
| DI-Attn | $85.78 \pm 0.13$ | $0.3222 \pm 0.0048$ | $99.07 \pm 0.47$ | $0.0245 \pm 0.0137$ |
| **AH (Ours)** | $\mathbf{89.29 \pm 0.16}$ | $\mathbf{0.2579 \pm 0.0017}$ | $\mathbf{99.39 \pm 0.11}$ | $\mathbf{0.0209 \pm 0.0037}$ |
| *Trigger: "movie"* | | | | |
| SI | $73.49 \pm 5.49$ | $0.5551 \pm 0.0382$ | $15.37 \pm 9.07$ | $1.3801 \pm 0.3992$ |
| DI-Std | $68.83 \pm 4.85$ | $0.5876 \pm 0.0309$ | $98.46 \pm 2.23$ | $0.1578 \pm 0.0654$ |
| DI-Attn | $88.72 \pm 0.55$ | $0.2902 \pm 0.0034$ | $97.66 \pm 0.64$ | $0.0612 \pm 0.0140$ |
| **AH (Ours)** | $\mathbf{89.20 \pm 0.46}$ | $\mathbf{0.2686 \pm 0.0059}$ | $\mathbf{99.30 \pm 0.61}$ | $\mathbf{0.0250 \pm 0.0136}$ |
| *Trigger: "the"* | | | | |
| SI | $75.99 \pm 4.03$ | $0.5399 \pm 0.0129$ | $13.88 \pm 6.88$ | $1.2502 \pm 0.3492$ |
| DI-Std | $64.77 \pm 5.99$ | $0.6450 \pm 0.0894$ | $\mathbf{84.95 \pm 13.02}$ | $0.4112 \pm 0.1301$ |
| DI-Attn | $\mathbf{87.59 \pm 0.54}$ | $\mathbf{0.3019 \pm 0.0064}$ | $83.60 \pm 2.96$ | $0.4094 \pm 0.0862$ |
| **AH (Ours)** | $86.93 \pm 0.41$ | $0.3028 \pm 0.0040$ | $84.16 \pm 1.30$ | $\mathbf{0.3758 \pm 0.0345}$ |

reported results were obtained using mixed-precision or full-precision training as specified in the configuration table.

**Model Initialization Protocol.** For both the distillation and evaluation stages, we utilize the official pre-trained weights (e.g., `bert-base-uncased`, `bert-tiny`) to leverage the linguistic priors of Transformers. Specifically, we load the pre-trained backbone weights but randomly re-initialize the classification head to ensure that the task-specific performance is derived solely from the distilled data rather than the pre-trained classifier.

**Label Initialization.** To strictly follow the standard initialization protocols, we employ *learnable soft labels* for all optimization-based methods (Clean-Std, Clean-Attn, DI, and AH) to allow gradient flow. In contrast, for the Static Injection (SI) baseline, we utilize fixed *hard labels* (one-hot vectors) to represent a naive, non-optimized poisoning approach.

**Attention Loss Definition.** To ensure reproducibility, we explicitly define the structural loss term used in Section 3.1. The attention loss $\mathcal{L}_{attn}$ is computed as the Kullback-Leibler (KL) divergence from the distilled labels $\tilde{a}$ to the model's attention maps $a(\theta)$:

$$\mathcal{L}_{attn} = D_{KL}(\tilde{a}||a(\theta)) = \sum \tilde{a} \log \frac{\tilde{a}}{a(\theta)}. \tag{7}$$

This direction forces the model's attention distribution to cover the modes defined by the distilled labels.

*Table 5.* **Detailed Results on AG News.** Comprehensive comparison of CTA and ASR metrics (reporting both Accuracy and Loss). **Bold** indicates the best performance (i.e., highest Accuracy and lowest Loss). Results marked with † denote utility collapse (i.e., CTA significantly degrading to unusable levels compared to the clean reference), and are thus excluded from the best-performance ranking despite high ASR.

| Method | Clean Test Accuracy (CTA) | | Attack Success Rate (ASR) | |
|---|---|---|---|---|
| | Accuracy (%) ↑ | Loss ↓ | Accuracy (%) ↑ | Loss ↓ |
| *Clean References* | | | | |
| Clean-Std | $65.97 \pm 2.09$ | $0.7488 \pm 0.0686$ | - | - |
| Clean-Attn | $\mathbf{92.03 \pm 0.08}$ | $\mathbf{0.2303 \pm 0.0014}$ | - | - |
| *Trigger: "cf"* | | | | |
| SI | $81.34 \pm 3.68$ | $0.5422 \pm 0.0828$ | $2.39 \pm 0.74$ | $3.8615 \pm 0.2755$ |
| DI-Std | $40.95 \pm 11.42^{\dagger}$ | $1.2618 \pm 0.2683^{\dagger}$ | $89.92 \pm 10.52^{\dagger}$ | $0.3938 \pm 0.2332^{\dagger}$ |
| DI-Attn | $\mathbf{91.67 \pm 0.09}$ | $\mathbf{0.2381 \pm 0.0014}$ | $\mathbf{99.97 \pm 0.01}$ | $\mathbf{0.0011 \pm 0.0006}$ |
| **AH (Ours)** | $91.63 \pm 0.08$ | $0.2480 \pm 0.0021$ | $99.80 \pm 0.04$ | $0.0075 \pm 0.0013$ |
| *Trigger: "said"* | | | | |
| SI | $82.47 \pm 1.85$ | $0.5316 \pm 0.0418$ | $2.98 \pm 1.53$ | $3.4822 \pm 0.4111$ |
| DI-Std | $59.84 \pm 5.66$ | $0.8509 \pm 0.1169$ | $95.08 \pm 9.14$ | $0.1867 \pm 0.2974$ |
| DI-Attn | $\mathbf{91.84 \pm 0.11}$ | $\mathbf{0.2359 \pm 0.0028}$ | $99.56 \pm 0.08$ | $0.0193 \pm 0.0049$ |
| **AH (Ours)** | $91.08 \pm 0.23$ | $0.2584 \pm 0.0052$ | $\mathbf{99.70 \pm 0.03}$ | $\mathbf{0.0110 \pm 0.0012}$ |
| *Trigger: "the"* | | | | |
| SI | $82.33 \pm 2.34$ | $0.5220 \pm 0.0737$ | $2.36 \pm 0.64$ | $4.1547 \pm 0.5763$ |
| DI-Std | $43.44 \pm 3.96^{\dagger}$ | $1.1647 \pm 0.0723^{\dagger}$ | $82.21 \pm 14.16^{\dagger}$ | $0.5607 \pm 0.3339^{\dagger}$ |
| DI-Attn | $\mathbf{91.89 \pm 0.05}$ | $\mathbf{0.2402 \pm 0.0005}$ | $43.05 \pm 0.77$ | $3.4281 \pm 0.2086$ |
| **AH (Ours)** | $91.53 \pm 0.17$ | $0.2466 \pm 0.0060$ | $\mathbf{62.53 \pm 2.35}$ | $\mathbf{1.6961 \pm 0.1266}$ |

## C. Overfitting and Optimization Dynamics

In this section, we provide a deeper investigation into the optimization dynamics of attention-guided dataset distillation, specifically addressing proxy overfitting under extreme data scarcity and the mechanisms by which AH stabilizes feature learning under severe linguistic interference.

### C.1. Mitigating Proxy Overfitting at Extreme Compression

Dataset Distillation at $SPC = 1$ inherently forces the proxy model to learn from an extremely sparse synthetic distribution, making it highly susceptible to overfitting. To rigorously investigate this phenomenon and evaluate potential mitigations, we explored both algorithmic regularization and data-scale expansion on the SST-2 dataset.

**Failure of Algorithmic Mitigation (Standard Regularization).** We first attempted to apply standard regularizers to the bi-level optimization process. Specifically, we enabled Dropout ($p = 0.1$) on the proxy model during the inner-loop and applied Weight Decay (0.01) to the synthetic data embeddings during the outer-loop update.

As shown in Table 8, applying these algorithmic mitigations catastrophically destroys the underlying DD process. Because the inner-loop is forced to unroll trajectories from exactly one sample per class, introducing stochasticity (Dropout) or weight penalties fundamentally shatters the deterministic computation graph required for precise gradient matching. The CTA collapses to $\sim 51\%$ (equivalent to random guessing), and the extreme variance in ASR ($\pm 38.98\%$) indicates that the stable backdoor injection dynamics are entirely disrupted.

**Data-Scale Mitigation.** Since algorithmic regularizers fail structurally, we mitigated the data scarcity bottleneck by scaling the compression rate to $SPC = 2$ and $SPC = 5$.

The joint analysis in Table 9 yields two critical insights. First, scaling successfully mitigates clean reference overfitting: expanding from $SPC = 1$ to $SPC = 2$ significantly improves the Clean-Attn's CTA from $85.02\%$ to $88.67\%$. Further scaling to $SPC = 5$ shows a marginal fluctuation, indicating that the proxy model reaches a performance saturation point

*Table 6.* **Ablation Study Numerical Results.** Detailed comparison of AH versus DI-Attn under varying hyperparameters and settings. `BERT-Base` is the default backbone unless otherwise specified.

| Experimental Setting | Method | Clean Test Accuracy (CTA) | | Attack Success Rate (ASR) | |
|---|---|---|---|---|---|
| | | Accuracy (%) ↑ | Loss ↓ | Accuracy (%) ↑ | Loss ↓ |
| *(a) Effect of Distillation Budget ($SPC$)* | | | | | |
| $SPC = 1$ (Default) | DI-Attn | $85.78 \pm 0.13$ | $0.3222 \pm 0.0048$ | $99.07 \pm 0.47$ | $0.0245 \pm 0.0137$ |
| | AH | $\mathbf{89.29 \pm 0.16}$ | $\mathbf{0.2579 \pm 0.0017}$ | $\mathbf{99.39 \pm 0.11}$ | $\mathbf{0.0209 \pm 0.0037}$ |
| $SPC = 2$ | DI-Attn | $87.98 \pm 0.28$ | $0.2831 \pm 0.0025$ | $94.02 \pm 1.79$ | $0.1243 \pm 0.0271$ |
| | AH | $\mathbf{88.62 \pm 0.22}$ | $\mathbf{0.2731 \pm 0.0034}$ | $\mathbf{98.74 \pm 0.32}$ | $\mathbf{0.0400 \pm 0.0093}$ |
| $SPC = 5$ | DI-Attn | $87.87 \pm 0.42$ | $0.2879 \pm 0.0052$ | $\mathbf{98.69 \pm 1.51}$ | $\mathbf{0.0367 \pm 0.0286}$ |
| | AH | $\mathbf{89.11 \pm 0.73}$ | $\mathbf{0.2652 \pm 0.0074}$ | $98.32 \pm 0.27$ | $0.0502 \pm 0.0043$ |
| *(b) Sensitivity to Attention Loss Weight ($\gamma$)* | | | | | |
| $\gamma = 0.5$ | DI-Attn | $86.10 \pm 0.43$ | $0.3272 \pm 0.0100$ | $97.10 \pm 1.09$ | $0.0632 \pm 0.0148$ |
| | AH | $\mathbf{88.17 \pm 0.20}$ | $\mathbf{0.2833 \pm 0.0040}$ | $\mathbf{99.21 \pm 0.46}$ | $\mathbf{0.0277 \pm 0.0126}$ |
| $\gamma = 1.0$ (Default) | DI-Attn | $85.78 \pm 0.13$ | $0.3222 \pm 0.0048$ | $99.07 \pm 0.47$ | $0.0245 \pm 0.0137$ |
| | AH | $\mathbf{89.29 \pm 0.16}$ | $\mathbf{0.2579 \pm 0.0017}$ | $\mathbf{99.39 \pm 0.11}$ | $\mathbf{0.0209 \pm 0.0037}$ |
| $\gamma = 2.0$ | DI-Attn | $\mathbf{88.42 \pm 0.48}$ | $\mathbf{0.2834 \pm 0.0077}$ | $99.25 \pm 0.09$ | $\mathbf{0.0388 \pm 0.0043}$ |
| | AH | $85.14 \pm 1.61$ | $0.3348 \pm 0.0207$ | $\mathbf{99.91 \pm 0.19}$ | $0.0404 \pm 0.0311$ |
| *(c) Impact of Poisoning Ratio ($\epsilon$)* | | | | | |
| $\epsilon = 0.1\%$ (Default) | DI-Attn | $85.78 \pm 0.13$ | $0.3222 \pm 0.0048$ | $99.07 \pm 0.47$ | $0.0245 \pm 0.0137$ |
| | AH | $\mathbf{89.29 \pm 0.16}$ | $\mathbf{0.2579 \pm 0.0017}$ | $\mathbf{99.39 \pm 0.11}$ | $\mathbf{0.0209 \pm 0.0037}$ |
| $\epsilon = 0.5\%$ | DI-Attn | $86.72 \pm 0.35$ | $0.3031 \pm 0.0045$ | $\mathbf{100.00 \pm 0.00}$ | $0.0020 \pm 0.0010$ |
| | AH | $\mathbf{87.25 \pm 0.17}$ | $\mathbf{0.2963 \pm 0.0050}$ | $\mathbf{100.00 \pm 0.00}$ | $\mathbf{0.0010 \pm 0.0007}$ |
| $\epsilon = 1.0\%$ | DI-Attn | $87.34 \pm 0.41$ | $0.3140 \pm 0.0045$ | $\mathbf{100.00 \pm 0.00}$ | $0.0016 \pm 0.0004$ |
| | AH | $\mathbf{88.00 \pm 0.33}$ | $\mathbf{0.3006 \pm 0.0053}$ | $\mathbf{100.00 \pm 0.00}$ | $\mathbf{0.0006 \pm 0.0003}$ |
| *(d) Model Scalability* | | | | | |
| `BERT-Tiny` | DI-Attn | $75.53 \pm 2.73$ | $0.5188 \pm 0.0320$ | $95.33 \pm 1.15$ | $0.0790 \pm 0.0158$ |
| | AH | $\mathbf{76.24 \pm 0.63}$ | $\mathbf{0.4810 \pm 0.0110}$ | $\mathbf{99.30 \pm 0.00}$ | $\mathbf{0.0076 \pm 0.0004}$ |
| `BERT-L6` | DI-Attn | $87.43 \pm 0.09$ | $0.3221 \pm 0.0017$ | $94.95 \pm 1.67$ | $\mathbf{0.1129 \pm 0.0318}$ |
| | AH | $\mathbf{88.44 \pm 0.27}$ | $\mathbf{0.2997 \pm 0.0083}$ | $\mathbf{95.00 \pm 1.33}$ | $0.1243 \pm 0.0204$ |
| `BERT-Base` (Default) | DI-Attn | $85.78 \pm 0.13$ | $0.3222 \pm 0.0048$ | $99.07 \pm 0.47$ | $0.0245 \pm 0.0137$ |
| | AH | $\mathbf{89.29 \pm 0.16}$ | $\mathbf{0.2579 \pm 0.0017}$ | $\mathbf{99.39 \pm 0.11}$ | $\mathbf{0.0209 \pm 0.0037}$ |

where information gain balances with bi-level optimization complexity. Second, the semantic anchor acts as an optimization stabilizer at extreme compression. At $SPC = 1$, where the Clean reference struggles heavily with overfitting, the AH semantic trigger ("film") naturally anchors the bi-level optimization trajectory along primary task features, allowing the model to maintain an optimal CTA of $89.29\%$. Most importantly, explicit data-level scaling does not wash out the backdoor, validating the persistent structural threat of AH.

## C.2. Structural Stabilizer and Feature Decoupling under High-Interference

In Section 4.2, we observed a striking phenomenon on the AG News dataset under the High-Interference setting (trigger: "the", appearing in $81.2\%$ of samples). While all baseline attacks suffered catastrophic utility and ASR collapse, AH uniquely maintained a potent attack without sacrificing CTA. This stability allows us to deduce the underlying mechanism by which AH functions as a structural stabilizer.

When a trigger is statistically indistinguishable from the most frequent linguistic background noise, traditional dynamic injection (e.g., DI-Attn) fails because the gradient signals of the primary task and the backdoor severely entangle, confusing the attention mechanism. AH fundamentally circumvents this by transforming the attention map from a flexible optimization

*Table 7.* **Hyperparameter Specifications.** Detailed configurations for the experiments. Specifically, the Inner Loop and Evaluation stages utilize SGD updates with a learnable learning rate derived from the distillation process. Note that the Attention Loss Weight ($\gamma$) applies to all attention-guided experiments (including clean references).

| General & Model Settings | |
|---|---|
| Backbone Architecture | `bert-base-uncased` |
| Maximum Sequence Length | 512 |
| Random Seed | 42 |
| Precision | FP32 (Full Precision) / Mixed Precision |
| **Data Distillation (Optimization & Budget)** | |
| Outer Optimizer | AdamW |
| Synthetic Data Learning Rate ($\eta_{\tilde{x}}$) | $1e-1$ (0.1) |
| Soft Label Learning Rate | $1e-1$ (0.1) |
| Inner Loop Optimizer (Proxy) | SGD |
| Inner Loop Learning Rate Init ($\eta_{\tilde{\theta}}$) | $1e-2$ (0.01, Learnable) |
| Inner Loop Steps ($M$) | 1 step per outer iteration |
| Distillation Epochs | 10 (based on Clean Data traversal) |
| Distillation Budget ($SPC$) | 1, 2, 5 (Default: 1) |
| Weight Decay | 0.0 |
| Warmup Ratio | 0.1 |
| Attention Loss Weight ($\gamma$) | 0.5, 1.0, 2.0 (Default: 1.0) |
| *Attack Configuration* | |
| Poisoning Ratio ($\epsilon$) | 0.1%, 0.5%, 1% (Default: 0.1%) |
| Trigger Position | Prefix (Index 1) |
| Attack Weight ($\lambda$) | 1.0 |
| Scaling Factor ($\alpha$) | 20 |
| **Final Evaluation (Testing)** | |
| Repeats | 5 independent runs |
| Optimizer | SGD |
| Learning Rate | Learned from Distillation |
| Batch Size | Matches SPC (Full Batch) |
| Validation Interval | Every 1 Epoch |

*Table 8.* **Impact of Algorithmic Mitigation on Utility and Attack ($SPC = 1$).** Standard regularizers catastrophically shatter the bi-level computation graph, resulting in utility collapse for both clean references and AH.

| Setup | Mitigation | Clean Test Accuracy (CTA) | | Attack Success Rate (ASR) | |
|---|---|---|---|---|---|
| | | Accuracy (%) ↑ | Loss ↓ | Accuracy (%) ↑ | Loss ↓ |
| Clean-Attn | None (Default) | $85.02 \pm 0.59$ | $0.3237 \pm 0.010$ | - | - |
| | Dropout + WD | $51.24 \pm 0.44$ | $0.7187 \pm 0.030$ | - | - |
| AH ("film") | None (Default) | $89.29 \pm 0.16$ | $0.2579 \pm 0.001$ | $99.39 \pm 0.11$ | $0.0209 \pm 0.003$ |
| | Dropout + WD | $51.28 \pm 2.59$ | $0.7050 \pm 0.010$ | $78.22 \pm 38.98$ | $0.6668 \pm 0.190$ |

*Table 9.* **Impact of Data-Scale Mitigation.** Expanding the data budget successfully mitigates clean reference overfitting up to a saturation plateau, while AH remains robust independent of the scale.

| Setup | Data Budget | Clean Test Accuracy (CTA) | | Attack Success Rate (ASR) | |
|---|---|---|---|---|---|
| | | Accuracy (%) ↑ | Loss ↓ | Accuracy (%) ↑ | Loss ↓ |
| Clean-Attn | $SPC = 1$ | $85.02 \pm 0.59$ | $0.3237 \pm 0.010$ | - | - |
| | $SPC = 2$ | $88.67 \pm 0.24$ | $0.2881 \pm 0.000$ | - | - |
| | $SPC = 5$ | $88.28 \pm 0.62$ | $0.2931 \pm 0.000$ | - | - |
| AH ("film") | $SPC = 1$ | $89.29 \pm 0.16$ | $0.2579 \pm 0.000$ | $99.39 \pm 0.11$ | $0.0209 \pm 0.000$ |
| | $SPC = 2$ | $88.62 \pm 0.22$ | $0.2731 \pm 0.000$ | $98.74 \pm 0.32$ | $0.0400 \pm 0.010$ |
| | $SPC = 5$ | $89.11 \pm 0.73$ | $0.2652 \pm 0.010$ | $98.32 \pm 0.27$ | $0.0502 \pm 0.010$ |

target into a hard, invariant constraint.

By explicitly forcing the global aggregator (`[CLS]` token) to lock its attention peak onto the noise trigger, AH prevents the model from attempting to extract meaningful primary task features from the trigger token itself. We empirically hypothesize that this explicit structural lock compels the network to execute an implicit feature decoupling: the model is forced to extract the clean task-essential features through the contextual embeddings of the remaining benign tokens. Consequently, the primary semantic features and the backdoor signal are spatially segregated into distinct representation pathways, avoiding catastrophic entanglement with the pervasive noise. This decoupling behavior explains why AH maintains high clean generalization (CTA) even when the attack signal relies on the dataset's most ubiquitous stopword.

## D. Detailed Analysis of Post-hoc and Structural Defenses

In addition to the empirical evaluations presented in Section 4.4, we provide a detailed theoretical and mechanistic analysis of advanced structural and post-training defense mechanisms. We argue that these canonical defenses either yield prohibitive false-positive rates, destroy task utility, or are fundamentally incompatible with the "Dataset-as-a-Service" threat model.

### D.1. Structural Defenses: Entropy and Smoothing

**Entropy-Based Anomaly Detection.**  While monitoring attention entropy is a conceptually intuitive defense against attention manipulation, it yields prohibitively high false-positive rates in the context of DD. Clean synthetic data's distilled attention labels naturally exhibit highly peaked, low-entropy distributions. This is a known structural byproduct of extreme information compression, which forces the proxy model to concentrate semantic information into key tokens. Consequently, applying absolute entropy thresholds would falsely flag the vast majority of benign distilled samples as poisoned, rendering the filtering mechanism impractical.

**Inference-Time Attention Smoothing.**  If a victim attempts to sanitize the model by artificially smoothing the attention scores at inference time (or post-hoc injecting noise into the downloaded attention labels), they introduce a severe distribution shift. In attention-guided DD frameworks, the proxy model is explicitly optimized to aggregate core features through quasi-one-hot attention peaks. The model's parameters become strictly coupled to this highly peaked attention distribution. Smoothing these scores disrupts the precise structural guidance required for knowledge transfer, decoupling the structural logic from the synthetic embeddings. This inevitably leads to a catastrophic collapse in CTA, imposing an unacceptable trade-off between security and primary task utility.

**Robust Distillation Regularizers.**  Applying structural constraints (e.g., uniformity or sparsity penalties) during the bi-level optimization process is fundamentally incompatible with our threat model. Under the "Dataset-as-a-Service" paradigm, the attacker completely controls and executes the compute-intensive distillation process. The victim merely downloads the resulting distilled datasets and lacks any access or authority to enforce robust regularizers during the upstream bi-level optimization phase.

### D.2. Advanced Post-Training Defenses

**Neural Cleanse.**  Canonical post-training defenses like Neural Cleanse attempt to reverse-engineer hidden triggers via optimization. However, this mechanism fails on extremely compressed distilled datasets. The severe lack of intra-class variance at $SPC = 1$ renders trigger reconstruction mathematically unstable, causing the defense optimization process to inevitably overfit to sample-specific features rather than isolating a universal backdoor pattern.

**Fine-Pruning.**  Defenses based on analyzing activation overlaps and pruning hijacked neurons typically require a substantially large, clean reference dataset. This strictly contradicts the foundational premise of DD usage, where resource-constrained users rely on synthetic data precisely because they lack access to massive proprietary data. Furthermore, because AH utilizes Semantic Anchoring to actively couple the trigger with robust benign features, forcibly pruning hijacked neurons fundamentally disrupts the coupled feature representations, catastrophically degrading the primary CTA.

