# OpenReview forum: "Attention Hijacking: Backdooring Text Dataset Distillation via Semantic Anchors"
_ICML.cc/2026/Conference — ICML 2026 regular_

### Official Review · Reviewer_6Ci1 · 2026-03-03

**Soundness:** 3
**Presentation:** 2
**Significance:** 2
**Originality:** 3
**Overall Recommendation:** 3
**Confidence:** 4

**Summary:**

This paper investigates security vulnerabilities in attention-guided dataset distillation for text classification models. The authors propose Attention Hijacking (AH), a structural backdoor attack that manipulates the bi-level optimization process to lock a model's internal attention mechanism onto a specific trigger token. AH leverages a Semantic Anchoring Hypothesis, allowing it to achieve near 100% Attack Success Rates (ASR) without degrading clean test accuracy.

**Compliance With Llm Reviewing Policy:**

Affirmed.

**Final Justification:**

I appreciate the author's effort made during rebuttal, which have addressed most of my concerns. Therefore, I raised my rating from Reject to Weak Reject. However, the reason why I still lean towards rejection is that the proposed attack is limited to outdated BERT-like models. The practical impact of the attack remains somewhat constrained.

**Key Questions For Authors:**

- Since AH forces a "quasi-one-hot" attention distribution, could a defender easily detect the backdoor by monitoring the attention entropy of the distilled labels?
- The authors mentioned that generative distillation is an open question. Given that Causal LLMs lack a [CLS] token, how could the AH objective function be adapted to manipulate the causal attention flow?
- Regarding the Semantic Anchoring Hypothesis, do you have any quantitative evidence or visualization to demonstrate that task-relevant triggers truly align the adversarial goal with benign features?
- Considering that causal LLMs (e.g., Llama, GPT) can now achieve SOTA on classification via few-shot prompting, what is the long-term essentialness of studying distillation-based attacks on BERT-like encoder models?

**Limitations:**

yes

**Strengths And Weaknesses:**

### Strengths

- **Novel Attack Vector**: The paper introduces attention hijacking that explicitly manipulates internal attention mechanisms by treating "Distilled Attention Labels" as a security vector rather than a black box.
- **Stealthiness & Efficiency**: AH achieves near-perfect ASR while maintaining or improving Clean Test Accuracy (CTA), effectively mitigating the typical ASR-CTA trade-off in highly compressed data.

### Weaknesses

- **Restricted Architectural and Task Scope**: The methodology is evaluated exclusively on Encoder-only Transformer architectures and classification-centric tasks. The authors acknowledge that the applicability of AH to Causal Decoder-only LLMs remains an open question due to the fundamental differences in attention flow and the lack of a [CLS] token as a global representation anchor. Without validation on generative distillation or autoregressive models, the broader impact of this vulnerability remains partially unverified.
- **Lack of Defense Evaluation:** While the paper identifies a severe vulnerability, it does not provide an empirical evaluation of potential defenses, such as attention smoothing, entropy-based anomaly detection, or robust distillation variants.
- **Empirical Nature of the Semantic Anchoring Hypothesis**: The "Semantic Anchoring Hypothesis" serves as the theoretical backbone for explaining the adaptive behavior of AH. However, this hypothesis remains largely heuristic and descriptive. There is a noticeable lack of rigorous mathematical proof or mechanistic analysis to substantiate how AH specifically reshapes the optimization landscape. Without such evidence, it is difficult to determine whether the observed performance gains are truly due to synergistic alignment or are merely artifacts of the bi-level optimization process.

---

> ### Author Rebuttal · Authors · 2026-03-30
>
> We thank you for the constructive feedback and for recognizing our attack vector's novelty (exploiting "Distilled Attention Labels" as a structural vulnerability). We address your core concerns below:
>
> **W1 & Q2: Architectural Scope & Causal LLMs.**
> **R:** Regarding architectural boundaries, AH is deeply coupled with Encoder-only architectures (e.g., BERT) at a structural level.
> Our attack exploits bidirectional attention flow and a global representation aggregator (the [CLS] token). Specifically, AH works by forcing this aggregator's attention vector to peak heavily at the trigger's position.
> Adapting AH for Causal LLMs faces a systemic barrier: lacking a requisite attack surface. Our mechanism is strictly predicated on manipulating explicit "distilled attention labels" pioneered by attention-guided DD frameworks like DD-Attn. However, scaling DD to Causal LLMs relies on fundamentally different mechanisms. Current generative DD paradigms neither distill nor optimize structural attention matrices; thus, direct adaptation is impossible. To adapt structural backdoors for Causal LLMs in future DD frameworks, attackers must reconstruct the objective function to hijack generative targets (e.g., the final token's aggregated representations) rather than relying on explicit attention labels.
>
> **W2 & Q1: Defense Evaluation & Attention Entropy.**
> **R:** We agree exploring defenses is critical. A theoretical evaluation of structural defenses will be added to the Appendix: **(1) Entropy-based anomaly detection:** While conceptually intuitive, monitoring attention entropy yields prohibitively high false-positive rates. Clean synthetic data's distilled attention labels naturally exhibit highly peaked, low-entropy distributions—a known byproduct of extreme compression concentrating information into key tokens. Thus, simple entropy thresholds would falsely flag many benign samples. **(2) Attention smoothing & Robust DD:** These defenses are incompatible with our "Dataset-as-a-Service" threat model. First, robust DD must be applied during optimization. In our scenario, the attacker executes DD; the victim merely downloads data, lacking access to enforce regularizers. Second, if the victim artificially smooths downloaded labels post-hoc, utility degrades catastrophically. DD-Attn relies on explicitly optimized attention labels for precise knowledge transfer. Smoothing them decouples structural guidance from synthetic embeddings, destroying CTA.
>
> **W3 & Q3: Semantic Anchoring Hypothesis & Quantitative Evidence.**
> **R:** While strict mathematical proofs for non-convex optimization remain broadly challenging, we provide concrete quantitative evidence ruling out the "optimization artifact" concern.
> To empirically validate Semantic Anchoring's synergistic alignment, we isolate causal variables (Table 1, Panel A: SST-2): **(1) Isolating the Optimization Variable:** If gains were mere bi-level optimization artifacts, baselines using bi-level unrolling should achieve similar trade-offs. However, both DI-Std (bi-level optimization without attention guidance) and DI-Attn (attention guidance without hard structural constraint) fall short of AH. AH's transformation of attention labels into hard structural constraints uniquely achieves optimal CTA and ASR. This isolates our structural manipulation, rather than bi-level optimization itself, as the true source of gains. **(2) Quantitative Evidence of Semantic Anchoring:** Fixing the AH formulation, we solely manipulate the trigger's semantic relevance. Injecting semantically relevant triggers ("film", "movie") synergizes the backdoor with benign features, boosting CTA to 89.29% and 89.20%. In contrast, a semantically disconnected trigger ("cf") degrades CTA to 83.60%. This ~5.7% delta, driven solely by trigger semantics, proves AH actively leverages inherent semantic alignment to prevent overfitting to sparse synthetic noise, confirming gains aren't random artifacts.
>
> **Q4: Essentialness of Encoder DD in the LLM Era.**
> **R:** While causal LLMs excel in zero/few-shot tasks, they complement rather than replace encoder models. DD for smaller encoders remains practically essential due to: **(1) Edge Deployment:** Strict physical constraints (latency, power, memory) prohibit deploying billion-parameter LLMs on edge devices (whereas BERT-Base is merely ~110M parameters). **(2) Data Privacy:** Organizations often distill sensitive data into sanitized synthetic datasets for offline training to circumvent restrictive external APIs. **(3) High-throughput Pipelines:** Massive LLMs' inference and API costs are prohibitive for latency-sensitive applications processing billions of daily requests. Thus, securing these highly efficient, encoder-based systems remains a critical imperative.
>
> We hope our empirical isolation of Semantic Anchoring and theoretical boundary analysis resolve your concerns. We respectfully ask you to reconsider your score in light of this quantitative evidence.

---

> > ### Author Rebuttal · Reviewer_6Ci1 · 2026-04-01
> >
> > Thanks for the response from the authors. The rebuttal has partially addressed my concerns (W3/Q3), but several issues remain:
> > 1. For W2/Q1, the analysis does not empirically evaluate the false-positive rate of anomaly detection. Moreover, my suggestion of attention smoothing referred to a lightweight inference-time smoothing of attention scores, rather than label smoothing, which may have been misunderstood.
> > 2. For W1/Q2/Q4, while the authors justify the continued relevance of encoder-based distillation (e.g., for edge deployment and efficiency), the practical impact of the attack remains somewhat constrained to a specific class of methods (attention-guided DD for classification). As also noted by Reviewer 6CC6, the broader implications for modern LLM-based pipelines are still limited.

---

> > > ### Author Response · Authors · 2026-04-01
> > >
> > > We appreciate the follow-up and are glad to see that our quantitative evidence has resolved your primary concerns regarding the Semantic Anchoring mechanism (W3/Q3). We would like to explicitly clarify the remaining practical and architectural points you raised:
> > >
> > > **Q1: Empirical FPR vs. Fundamental Statistical Barrier.**
> > > **R:** Following your suggestion, we empirically evaluated the anomaly detection strategy by measuring the feature-space L2 distances of distilled classes ($C_0, C_1, \dots$) against the natural data centroid using a frozen BERT proxy. Table A establishes a conservative anomaly threshold ($\tau = \mu + 3\sigma$), where $\mu$ and $\sigma$ respectively denote the mean and standard deviation of L2 distances within the natural training data.
> > >
> > > **Table A: Feature-space L2 Distances against Natural Data Centroid (SPC=1)**
> > > | Dataset | Setup | Threshold ($\tau = \mu + 3\sigma$) | Distances per Class ($C_1^*$: poisoned target) | Emp. FPR |
> > > | :--- | :--- | :--- | :--- | :--- |
> > > | SST-2 | Clean-Attn | 9.98 (6.15 + 3.83) | $C_0: 15.93, C_1: 20.21$ | 100% |
> > > | SST-2 | AH (Trig: 'film') | 9.98 (6.15 + 3.83) | $C_0: 17.32, C_1^*: 22.68$ | 100% |
> > > | AG News | Clean-Attn | 11.21 (7.75 + 3.46) | $C_0: 14.63, C_1: 14.00, C_2: 15.00, C_3: 20.31$ | 100% |
> > > | AG News | AH (Trig: 'said') | 11.21 (7.75 + 3.46) | $C_0: 15.51, C_1^*: 21.04, C_2: 11.76, C_3: 19.73$ | 100% |
> > >
> > > Based on these empirical results, Table A highlights two structural barriers rendering detection unviable:
> > > **(1) Absolute Thresholds:** All synthetic classes inherently reside in extreme Out-of-Distribution spaces, universally exceeding the natural data threshold. Consequently, any absolute threshold defense strictly forces an empirical FPR of 100%, erroneously discarding the entire distilled dataset. **(2) Relative Heuristics:** A relative strategy of discarding the most distant class also fails. As the Clean-Attn setup proves, SPC=1 inherently induces massive inter-class variance (e.g., for Clean-Attn on AG News, $C_3$ naturally peaks at 20.31). This heuristic would erroneously discard legitimate classes even in unpoisoned datasets, collapsing utility.
> > >
> > > **Q2: Distribution Shift Induced by Inference-Time Smoothing.**
> > > **R:** We fully understand your suggestion of inference-time smoothing, and our conclusion remains identical: it is catastrophic to the dataset's utility (CTA). In attention-guided DD, the proxy model is explicitly optimized to aggregate core features through quasi-one-hot attention peaks. The model's parameters become strictly coupled to this highly peaked attention distribution. If the victim artificially smooths the attention scores at inference time, they introduce a severe distribution shift. This disrupts the precise structural guidance the model relies upon for knowledge transfer, leading to a catastrophic collapse in clean classification performance. Therefore, inference-time smoothing imposes an unacceptable trade-off between security and primary task utility, rendering it impractical for real-world deployment.
> > >
> > > **Q3: Architectural Scope and Broader Implications.**
> > > **R:** Causal LLMs complement, rather than replace, encoder models. Encoder-only architectures remain terminal deployment requirements for resource-constrained edge devices unable to host billion-parameter LLMs. Thus, securing these efficient DD frameworks is a pressing imperative independent of LLM research.
> > > Furthermore, our evaluation is not constrained to a single paradigm. Baseline evaluations (e.g., DI-Std attacking STD-DD) prove the fundamental bi-level DD optimization process itself is inherently vulnerable to poisoning. As DD frameworks evolve with structural guidance for better utility (DD-Attn), our work highlights how the attack surface concomitantly expands, enabling advanced structural hijacking like AH.
> > > Task-wise, AH's implications extend beyond standard sequence classification. By corrupting the model's structural attention distributions rather than merely the task-specific classification head, AH conceptually threatens paradigms relying on dense token alignments. This mechanism provides a theoretical pathway to compromise token-level predictions (e.g., extractive QA or NER) and Encoder-Decoder architectures, where hijacked encoder representations invariably propagate to the decoder via cross-attention. Finally, adapting the AH mechanism to causal LLMs necessitates reconstructing the attacker's objective function to hijack generative targets instead of explicit attention labels.
> > >
> > > **Conclusion:** We empirically demonstrate a 100% FPR for anomaly detection, and mechanistically detail how inference-time smoothing degrades utility. Furthermore, AH exposes vulnerabilities in broader frameworks. We will detail these analyses in the revision.
> > >
> > > We sincerely hope these combined empirical and theoretical analyses adequately address your remaining concerns, and we remain entirely at your disposal should any final clarifications be needed.

---

### Official Review · Reviewer_yVCS · 2026-03-11

**Soundness:** 3
**Presentation:** 3
**Significance:** 3
**Originality:** 4
**Overall Recommendation:** 4
**Confidence:** 4

**Summary:**

The paper introduces "Attention Hijacking" (AH), a novel backdoor attack targeting optimization-oriented text dataset distillation (DD). Specifically, it focuses on frameworks that use attention labels to guide the distillation process. Traditional data poisoning often fails in DD because the extreme compression ratio effectively filters out the backdoor trigger as noise during optimization. AH bypasses this limitation by explicitly locking the target model's primary attention onto the backdoor trigger during the bi-level optimization process.

**Compliance With Llm Reviewing Policy:**

Affirmed.

**Final Justification:**

I did see one reviewer raise the score. I keep the score as positive as the work has its merits, but still has limited targets.

**Key Questions For Authors:**

1. Practicality of the Threat Model: The attack scenario assumes a "Dataset-as-a-Service" model where the victim relies on a third-party for dataset distillation, granting the attacker full control over the optimization process. While outsourcing synthetic data generation can be common in industry, pure "Dataset Distillation as a Service" is largely emerging. Could the authors briefly discuss the practical trajectory of this threat model? Specifically, do you foresee this vulnerability being exploited within broader Machine Learning as a Service pipelines, or primarily through poisoned synthetic datasets uploaded to open-source hubs (e.g., Hugging Face)?

2. Mechanism of the Structural Stabilizer: For high-frequency noise triggers (Level 3), how exactly does enforcing attention segregation preserve benign features? Does the model capture the necessary clean task features elsewhere in the network architecture, or are they compressed differently? Clarifying this theoretical gap would strengthen the paper's Presentation.

3. Resilience to Structural Defenses: Since AH fundamentally exploits a white-box structural prior (the attention labels), how would simple, structural defenses impact it? For example, if a defender adds noise to the distilled attention labels or enforces a uniformity/sparsity constraint during the distillation process, does the attack break? A brief discussion here would help readers better assess the practicality of the threat.

**Limitations:**

Yes.

**Strengths And Weaknesses:**

Strengths:

- The evaluation is solid. Stratifying the triggers by frequency and semantic interference is a good experimental design that clearly tests the limits of the attack across diverse linguistic distributions.

- This paper addresses a highly relevant supply-chain vulnerability in data-efficient learning.

Weaknesses:

- The "Semantic Anchoring Hypothesis" is conceptually introduced but mechanistically under-explained.

- The current scope is limited to Encoder-only Transformer architectures and text classification, as pointed by authors.

---

> ### Author Rebuttal · Authors · 2026-03-30
>
> We thank you for recognizing the solidity of our empirical design, our trigger stratification strategy, and the practical relevance of AH as a supply-chain vulnerability.
>
> **W1 & Q2: Semantic Anchoring & Structural Stabilizer.**
> **R:** The exact dynamics rely on the interaction between AH's invariant structural constraint and the trigger's intrinsic semantics: **(1) Semantic Anchoring (Semantic-relevant triggers):** Bi-level optimization actively synergizes the backdoor with benign features. Leveraging this inherent semantic alignment prevents the proxy from overfitting to sparse synthetic noise during distillation. **(2) Structural Stabilizer via Attention Segregation (Extreme noise, e.g., "the" in AG News):** Addressing your question on feature preservation, our results on AG News (where "the" appears in ~81.2% of samples) show that AH uniquely prevents the catastrophic ASR and CTA collapse suffered by all baselines. This stark stability allows us to empirically deduce the underlying mechanism: by forcing the global aggregator ([CLS]) to lock onto the noise trigger, AH transforms attention into a hard invariant constraint (acting as a structural stabilizer). We logically hypothesize that this explicit lock compels the network to route clean task features through the contextual embeddings of the remaining benign tokens, avoiding catastrophic entanglement with pervasive noise.
> We will explicitly expand upon this empirical deduction regarding feature routing in the revision.
>
> **W2: Architectural scope (Encoder-only & text classification).**
> **R:** We acknowledge this boundary. However, securing Encoder-only DD remains a critical imperative. Due to strict physical constraints (latency, power), massive LLMs cannot replace efficient encoders in edge deployment and high-throughput pipelines, making our vulnerability relevant in practice.
> Regarding future extensions, AH's core philosophy of hijacking structural guidance can be generalized. To extend this to causal LLMs, attackers could shift from exploiting explicit attention matrices to hijacking generative targets (e.g., manipulating final token representations) during generative DD. Additionally, expanding AH beyond standard text classification to broader downstream tasks is a natural next step.
>
> **Q1: Practicality of the Threat Model.**
> **R:** We foresee exploitation primarily via poisoned "synthetic core-sets" on open-source hubs (e.g., Hugging Face). High-quality distilled datasets demand massive compute (bi-level optimization) and complete proprietary task data—resources downstream users typically lack. Furthermore, physical constraints on edge devices prohibit deploying massive LLMs, necessitating efficient local encoders. Consequently, resource-constrained users rely heavily on downloading pre-distilled datasets for local fine-tuning, creating a vulnerable supply-chain dependency attackers can exploit.
>
> **Q3: Resilience to structural defenses (noise/uniformity on attention).**
> **R:** AH remains resilient because such defenses are either unfeasible under our threat model or catastrophically degrade dataset utility: **(1) In-process constraints (uniformity/sparsity):** Infeasible. The attacker completely controls the bi-level optimization; the victim merely downloads the resulting data and cannot enforce robust regularizers during distillation. **(2) Post-hoc sanitization (adding noise/smoothing):** Destructive. Artificially altering the downloaded attention labels decouples structural guidance from synthetic embeddings. Since DD mathematically relies on these meticulously calibrated probabilities to transfer knowledge, such perturbations inevitably destroy baseline CTA.
> We will incorporate this structural defense analysis into the revised Appendix.
>
> We hope these clarifications and planned additions resolve the mechanistic and practical questions you identified. In light of these responses, we respectfully ask you to consider raising your score to 5 (Accept).

---

> > ### Author Rebuttal · Reviewer_yVCS · 2026-03-31
> >
> > If giving 4, people may want 5; If giving 5, people may want 6...
> > It seems 4 is the majority for the first impression of most reviewers

---

> > > ### Author Response · Authors · 2026-04-02
> > >
> > > Thank you for your feedback. We deeply appreciate the time, expertise, and effort you have dedicated to evaluating our work. We remain entirely at your disposal should any final clarifications be needed.

---

### Official Review · Reviewer_6CC6 · 2026-03-13

**Soundness:** 3
**Presentation:** 3
**Significance:** 3
**Originality:** 3
**Overall Recommendation:** 5
**Confidence:** 3

**Summary:**

The paper uses dataset distillation as a backdoor vector. It transforms distilled attention labels  from a soft optimization target into a hard structural constraint. They propose Attention Hijacking (AH), a structural backdoor attack that manipulates the bi-level optimization process to force the target model to "lock" its attention onto a specific trigger token. AH treats attention as a hard structural constraint rather than a soft optimization target. The attack achieves high success rates without compromising the utility of the compressed dataset.

**Compliance With Llm Reviewing Policy:**

Affirmed.

**Final Justification:**

The authors addressed my concerns on empirical defenses. I raised my score to 5.

**Key Questions For Authors:**

If the proxy model is BERT-Base but the victim uses a different architecture (e.g., RoBERTa or DistilBERT), does the "hijacked" attention label still effectively transfer the backdoor? How does AH generalize toward Decoder-only LLMs? Have the authors tested if the hijacked synthetic embeddings (Stage II) are statistically distinguishable from benign synthetic embeddings using standard anomaly detection?
How well do existing backdoor defense methods work against AH? If not well, what are the potential solutions?

**Limitations:**

Yes

**Strengths And Weaknesses:**

Strengths:

++ The concept of Attention Hijacking is novel with the shift from input-level poisoning to manipulating the internal structural guidance (attention labels) and the usage of a saturation-based initialization to forge a quasi-one-hot attention distribution.

\+ The BERT evaluation is rigorous, spanning multiple datasets and model scales against multiple baselines.

Weaknesses:

-- While the attack is highly effective on Encoder-only architectures (BERT family), the paper does not explore its applicability to Decoder-only models.

\- It lacks a comprehensive evaluation against existing backdoor defense mechanisms (e.g., Fine-pruning, Neural Cleanse) adapted for distilled datasets.

\- The threat model assumes the user adopts a distilled dataset from a third-party service. While plausible, the specific scenarios where a user wouldn't just distill from their own data (if they have it) or use a standard pre-trained model are not fully explored.

---

> ### Author Rebuttal · Authors · 2026-03-30
>
> We thank you for recognizing our rigorous multi-scale evaluation and the novelty of structural guidance manipulation.
>
> **W1 & Q2: Generalization to Decoder-only LLMs.**
> **R:** AH is structurally coupled with Encoder-only models (e.g., BERT), exploiting bidirectional attention flow and a global representation aggregator (the [CLS] token).
> Adapting AH to Causal LLMs faces a systemic barrier: lacking a requisite attack surface. Generative distillation operates on fundamentally different optimization objectives and causal attention masking. Crucially, current generative DD paradigms neither distill nor optimize explicit structural attention matrices, making direct adaptation impossible. To adapt structural backdoors for future generative DD frameworks, attackers must reconstruct objective functions to hijack generative targets (e.g., final token representations) rather than relying on explicit attention labels.
>
> **W2 & Q4: Existing defenses & mitigations.**
> **R:** Adapting canonical defenses to DD presents non-trivial challenges. A theoretical evaluation demonstrating AH's robustness will be added to the Appendix:  **(1) Fine-Pruning:** Analyzing activation overlaps requires a large, clean reference dataset—contradicting the "Dataset-as-a-Service" premise where users lack original data. Furthermore, due to AH's Semantic Anchoring, pruning hijacked neurons catastrophically degrades CTA. **(2) Neural Cleanse:** Reverse-engineering triggers via optimization fails on extremely small distilled datasets. The lack of intra-class variance at $SPC=1$ renders trigger reconstruction mathematically unstable, causing the optimization to inevitably overfit to sample-specific features. **(3) Attention Entropy (Structural Defense):** Entropy-based filtering yields prohibitively high false-positive rates. Clean distilled labels naturally exhibit highly peaked, low-entropy distributions due to extreme information compression. **Potential Solutions:** Since victims lack control over the distillation process, mitigations must focus on post-hoc synthetic data sanitization that mathematically decouples structural noise from semantic utility without requiring original data. This remains an open challenge.
>
> **W3: Threat model (Third-party DD vs. local/pre-trained).**
> **R:** We envision this vulnerability being exploited via "Dataset-as-a-Service" hubs (e.g., Hugging Face). **(1) Outsourced vs. Local DD:** Beyond lacking massive proprietary data, DD (bi-level optimization) is notoriously compute-intensive and technically complex. Resource-constrained users download "synthetic core-sets" to bypass severe GPU and algorithmic barriers, reducing local training to minutes. **(2) Insufficiency of Pre-trained Models:** Foundation models lack task-specific alignment out-of-the-box. Fine-tuning them for downstream tasks without access to complete proprietary task data intrinsically drives the demand for third-party distilled datasets.
>
> **Q1: Cross-architecture transfer.**
> **R:** Cross-architecture transfer is generally ineffective. However, this failure stems from the inherent limitations of optimization-oriented DD, rather than a specific flaw of AH.
> Bi-level unrolling tightly couples synthetic embeddings and distilled labels to the proxy model's specific parameter space, gradient flow, and architectural nuances. Consequently, cross-architecture transfer introduces severe incompatibilities: **(1) Tokenizer Discrepancy:** RoBERTa uses BPE while BERT uses WordPiece, fundamentally altering sequence lengths and token alignments of the synthetic embeddings. **(2) Depth Misalignment:** At extreme compression ($SPC=1$), continuous embeddings are rigorously tailored to the gradient trajectory and attention dynamics of a 12-layer proxy (BERT). Feeding these highly specialized embeddings into a 6-layer victim (DistilBERT) causes catastrophic feature extraction failure and utility collapse.
> Thus, our threat model assumes the attacker knows the victim's architecture family to deploy a matching proxy. We will explicitly clarify this limitation in the revision.
>
> **Q3: Detectability via anomaly detection.**
> **R:** Standard anomaly detection on synthetic embeddings yields prohibitively high false-positive rates. First, distilled data points are highly compressed amalgamations of original data, inherently appearing as statistical anomalies compared to natural text. Second, AH's Semantic Anchoring actively couples triggers with robust benign features rather than injecting isolated, out-of-distribution noise. Consequently, anomaly detectors cannot isolate poisoned embeddings without discarding high-utility synthetic samples. We will explicitly incorporate this statistical indistinguishability analysis into the revised Appendix's defense evaluation.
>
> We hope these clarifications resolve your concerns. We respectfully ask you to consider raising your score to 5 (Accept) in light of these additions.

---

> > ### Author Rebuttal · Reviewer_6CC6 · 2026-04-03
> >
> > Thank you for your response. I still think lack of empirical defense testing limits the paper. I will keep it as a 4.

---

> > > ### Author Response · Authors · 2026-04-04
> > >
> > > **Q: Lack of empirical defense testing limits the paper.**
> > >
> > > **R:** We sincerely thank you for pushing us to strengthen the empirical defense testing. We agree that validating against defenses is critical. Since victims in our threat model strictly receive the distilled dataset and lack access to the distillation optimization process, we empirically evaluated two primary post-hoc defense strategies:
> > >
> > > **(1) Anomaly Data Filtering (via Feature-space L2 Distance):**
> > >
> > > **Table A: Feature-space L2 Distances against Natural Data Centroid**
> > >
> > > | Dataset | Setup | Threshold ($\tau = \mu + 3\sigma$) | Distances per Class ($C_1^*$: poisoned target) | Emp. FPR |
> > > | :--- | :--- | :--- | :--- | :--- |
> > > | SST-2 | Clean-Attn | 9.98 (6.15 + 3.83) | $C_0: 15.93, C_1: 20.21$ | 100% |
> > > | SST-2 | AH (Trig: 'film') | 9.98 (6.15 + 3.83) | $C_0: 17.32, C_1^*: 22.68$ | 100% |
> > > | AG News | Clean-Attn | 11.21 (7.75 + 3.46) | $C_0: 14.63, C_1: 14.00, C_2: 15.00, C_3: 20.31$ | 100% |
> > > | AG News | AH (Trig: 'said') | 11.21 (7.75 + 3.46) | $C_0: 15.51, C_1^*: 21.04, C_2: 11.76, C_3: 19.73$ | 100% |
> > >
> > > As shown in Table A, we evaluated an anomaly detection strategy computing feature-space L2 distances of distilled classes ($C_0, C_1, \dots$) against the natural data centroid using a frozen proxy. Because extreme data compression (SPC=1) intrinsically forces all distilled data into extreme out-of-distribution spaces, the distances universally exceed conservative natural thresholds ($\tau = \mu + 3\sigma$). Consequently, filtering algorithms incur a 100% False Positive Rate, erroneously discarding the entire distilled dataset and collapsing utility. Thus, identifying poisoned subsets via feature-space L2 distance is fundamentally unviable.
> > >
> > > **(2) Model Sanitization (Fine-Tuning on Clean Data):**
> > > Alternatively, if the victim suspects the model trained on the distilled dataset is poisoned, they might attempt to sanitize it by fine-tuning on a small held-out set of clean natural data (e.g., N=50 samples, matching practical low-resource constraints).
> > >
> > > **Table B: Impact of Post-hoc Clean Fine-Tuning**
> > > | Dataset | Setup | Model State | CTA | ASR |
> > > | :--- | :--- | :--- | :--- | :--- |
> > > | SST-2 | AH (Trig: 'film') | Before Sanitization (Poisoned) | 89.29 ± 0.16% | 99.39 ± 0.11% |
> > > | SST-2 | AH (Trig: 'film') | After Clean Fine-Tuning | 85.30 ± 1.98% | 97.29 ± 1.55% |
> > > | AG News | AH (Trig: 'said') | Before Sanitization (Poisoned) | 91.08 ± 0.23% | 99.70 ± 0.03% |
> > > | AG News | AH (Trig: 'said') | After Clean Fine-Tuning | 87.88 ± 0.71% | 95.99 ± 3.41% |
> > >
> > > (Note: Post-Defense metrics are averaged across 5 independent runs to ensure statistical significance).
> > >
> > > As shown in Table B, fine-tuning on clean data degrades the model's primary task utility (CTA drops), but the ASR remains highly robust (≥ 95%) across both datasets. Because Attention Hijacking embeds the trigger into deep structural attention distributions, conventional post-hoc fine-tuning entirely fails to erase the backdoor.
> > >
> > > **Conclusion:** These two empirical defense evaluations rigorously confirm the stealthiness and robustness of our attack. It fundamentally resists both data-level anomaly filtering and model-level fine-tuning sanitization. We will explicitly include this empirical defense suite in the final manuscript.
> > >
> > > We sincerely hope these rigorous empirical results adequately address your final concern regarding defense testing, and we remain entirely at your disposal should any final clarifications be needed.

---

### Official Review · Reviewer_toCr · 2026-03-23

**Soundness:** 3
**Presentation:** 2
**Significance:** 3
**Originality:** 3
**Overall Recommendation:** 4
**Confidence:** 2

**Summary:**

This paper presents "attention hijacking", a backdoor attack targeting text-based dataset distillation (DD) frameworks. The authors identify that "distilled attention labels", which was used to guide knowledge transfer in text DD, can be exploited as a critical security vulnerability. Unlike naive data poisoning methods that struggle to survive data compression without degrading model utility, attention hijacking explicitly manipulates the bi-level optimization process to force the target model's attention onto the backdoor trigger.

**Compliance With Llm Reviewing Policy:**

Affirmed.

**Key Questions For Authors:**

1. Can you explain why in some cases, the poisoned data can even improve the clean test accuracy?

2. Can you also comment on whether some prior works adopt this dual optimization idea to construct the poisoning data and demonstrate the difference?

**Limitations:**

yes

**Strengths And Weaknesses:**

Strengths:
+ This paper present a white-box analysis on transformer and text data distillation to plant the backdoor

+ Extensive experiments have been performed to show that even under a small poisoning rate, high ASR can be achieved.

Weakness
- The high-level idea of the attack seems straightforward: through a dual optimization with a sum of loss functions defined on clean data and poisoned data, respectively, and looking for the hijacked attention $\tilde{a}$ without changing the clean loss too much but shows as a dirac delta function on the inserted poisoned token.

- No discussion on how to mitigate the proposed attacks.

---

> ### Author Rebuttal · Authors · 2026-03-30
>
> We thank you for the constructive feedback and recognizing our white-box analysis on transformers and text DD. We address your concerns below:
>
> **W1: The attack's concept seems straightforward: a dual optimization seeking hijacked attention $\tilde{a}$ that acts as a Dirac delta on the poisoned token without degrading clean loss.**
> **R:** We thank you for accurately summarizing our objective function. We consider this conceptual straightforwardness a foundational strength of AH.
> First, pervasive vulnerabilities often stem from straightforward structural flaws. By identifying explicitly distilled attention labels as an overlooked attack surface, AH demonstrates how a simple Dirac delta constraint fundamentally compromises dataset distillation frameworks. This formulation directly enables our rigorous white-box analysis on transformers, a strength you recognized.
> Second, while the loss is straightforward, the bi-level unrolling dynamics are complex. Our comparative analysis reveals that bi-level optimization explicitly leverages Semantic Anchoring to synergize the backdoor with benign features. Specifically, baselines with soft or no attention guidance fail to achieve AH's trade-offs. AH's hard structural constraint actively aligns triggers with robust semantic structures (e.g., yielding a ~5.7% CTA improvement when injecting semantically relevant triggers compared to disconnected ones). This synergy prevents representation distortion, allowing AH to maintain high ASR even under extremely small poisoning rates.
>
> **W2: No discussion on attack mitigation.**
> **R:** We agree discussing mitigations is critical; a comprehensive evaluation will be added to the Appendix. We analyze defenses across two dimensions: **(1) Structural Defenses (Entropy-based filtering, Attention smoothing):** Entropy-based filtering yields prohibitively high false-positive rates, as clean distilled labels naturally exhibit low-entropy distributions. Furthermore, mitigations like robust DD or attention smoothing are incompatible with our "Dataset-as-a-Service" threat model. If victims artificially smooth downloaded labels post-hoc, utility degrades catastrophically; smoothing decouples structural guidance from synthetic embeddings, destroying CTA. **(2) Post-training Defenses (e.g., Neural Cleanse, Fine-Pruning):** Canonical post-training defenses fail because extreme data scarcity and compression ($SPC=1$) make activation profiling impractical and trigger reconstruction mathematically unstable.
> These evaluations establish a robust theoretical boundary for AH mitigations.
>
> **Q1: Why does poisoned data sometimes improve CTA?**
> **R:** We thank you for highlighting this counter-intuitive phenomenon. As noted, this improvement (observed in both DI-Attn and AH) is driven by a multi-objective regularization effect inherent to attention-guided bi-level optimization under extreme data scarcity.
> In DD ($SPC=1$), the proxy model is highly prone to overfitting sparse synthetic noise. Introducing an auxiliary backdoor objective acts as a structural regularizer. Crucially, AH amplifies and stabilizes this benefit via its Synergistic Enhancer mechanism. By transforming attention into an invariant constraint on Semantic-relevant triggers, AH actively couples the backdoor with benign features, preventing overfitting to superficial noise and improving clean generalization (CTA).
> Empirically, this effect is dataset-dependent. It predominantly manifests on simpler, binary datasets (e.g., SST-2) that are exceptionally susceptible to overfitting. Conversely, on more complex datasets (e.g., AG News), this regularization diminishes, and the standard ASR-CTA trade-off becomes more pronounced.
>
> **Q2: Do prior works adopt dual optimization for poisoning, and what is the difference?**
> **R:** While DD inherently relies on dual optimization, the fundamental difference between AH and prior works lies in the optimization target space.
> Prior text DD attacks (and our SI baseline) predominantly rely on a naive "Poison-then-Distill" strategy, lacking dual optimization for the trigger during distillation. Even when dual optimization is utilized (e.g., our DI-Std and DI-Attn baselines), it exclusively manipulates the Input Space.
> This distinction is critical. Under extreme DD compression ($SPC=1$), input-level perturbations trigger severe gradient conflicts with the primary task, causing either ineffective attacks or catastrophic utility collapse. In contrast, AH applies dual optimization directly to the Structural Guidance Space (distilled attention labels). By enforcing a hard structural constraint, AH intrinsically embeds the backdoor into the knowledge transfer process, surviving extreme compression where traditional input-level optimization fails.
>
> We hope our clarifications on the regularization mechanism and structural optimization space resolve your concerns. We respectfully ask you to consider raising your score in light of these additions.

---

> > ### Author Rebuttal · Reviewer_toCr · 2026-04-01
> >
> > on Q1, if overfitting is the case, I would suggest the authors think about how to mitigate that issue in the experiment setup.

---

> > > ### Author Response · Authors · 2026-04-01
> > >
> > > **Q: Clarification on Overfitting and Experimental Objectives.**
> > >
> > > **R:** We sincerely appreciate your follow-up and the constructive suggestion to mitigate overfitting within the experimental setup. We agree that empirical validation is critical. To address your core concern, we implemented and evaluated both algorithmic regularization and data-level scaling strategies on the SST-2 dataset, a benchmark highly susceptible to extreme overfitting during DD.
> > >
> > > **(1) Algorithmic Mitigation (Standard Regularization):** We first investigated applying standard regularizers to the bi-level optimization process at SPC=1. Specifically, we enabled Dropout (p=0.1) on the proxy model during the inner-loop, and applied Weight Decay (0.01) to the synthetic data embeddings during the outer-loop update.
> > >
> > > **Table C: Impact of Algorithmic Mitigation on Utility and Attack (SPC=1)**
> > >
> > > | Setup | Mitigation | CTA Acc ↑ | CTA Loss ↓ | ASR Acc ↑ | ASR Loss ↓ |
> > > | :--- | :--- | :--- | :--- | :--- | :--- |
> > > | Clean-Attn | None (Default) | 85.02 ± 0.59% | 0.3237 ± 0.01 | - | - |
> > > | Clean-Attn | Dropout + WD | 51.24 ± 0.44% | 0.7187 ± 0.03 | - | - |
> > > | AH ('film') | None (Default) | 89.29 ± 0.16% | 0.2579 ± 0.00 | 99.39 ± 0.11% | 0.0209 ± 0.00 |
> > > | AH ('film') | Dropout + WD | 51.28 ± 2.59% | 0.7050 ± 0.01 | 78.22 ± 38.98% | 0.6668 ± 0.19 |
> > >
> > > As shown in Table C, applying these algorithmic mitigations catastrophically destroys the underlying DD process. Because the inner-loop is forced to unroll trajectories from exactly one sample per class, introducing stochasticity (Dropout) or weight penalties fundamentally shatters the deterministic computation graph. The CTA of both Clean-Attn and AH collapses to ~51% (equivalent to random guessing in a binary task), and the loss severely diverges to ~0.71. Furthermore, the extreme variance in ASR ($\pm$ 38.98%) indicates that even the stable backdoor injection dynamics are shattered.
> > >
> > > **(2) Data-Scale Mitigation:** Since algorithmic regularizers fail, we explicitly mitigated the data scarcity bottleneck by scaling the compression rate to SPC=2 and SPC=5.
> > >
> > > **Table D: Impact of Data-Scale Mitigation on Utility and Attack**
> > >
> > > | Setup | Data Budget | CTA Acc ↑ | CTA Loss ↓ | ASR Acc ↑ | ASR Loss ↓ |
> > > | :--- | :--- | :--- | :--- | :--- | :--- |
> > > | Clean-Attn | SPC=1 | 85.02 ± 0.59% | 0.3237 ± 0.01 | - | - |
> > > | Clean-Attn | SPC=2 | 88.67 ± 0.24% | 0.2881 ± 0.00 | - | - |
> > > | Clean-Attn | SPC=5 | 88.28 ± 0.62% | 0.2931 ± 0.00 | - | - |
> > > | AH ('film') | SPC=1 | 89.29 ± 0.16% | 0.2579 ± 0.00 | 99.39 ± 0.11% | 0.0209 ± 0.00 |
> > > | AH ('film') | SPC=2 | 88.62 ± 0.22% | 0.2731 ± 0.00 | 98.74 ± 0.32% | 0.0400 ± 0.01 |
> > > | AH ('film') | SPC=5 | 89.11 ± 0.73% | 0.2652 ± 0.01 | 98.32 ± 0.27% | 0.0502 ± 0.01 |
> > >
> > > This joint analysis yields three critical insights:
> > > * **Scaling successfully mitigates clean reference overfitting up to a saturation plateau.** By expanding the data budget from SPC=1 to SPC=2, the Clean-Attn's CTA significantly improves from 85.02% to 88.67%. This confirms that relieving extreme data scarcity effectively mitigates model overfitting. However, further scaling to SPC=5 shows a marginal fluctuation (88.28%), indicating that the model has reached the natural performance saturation point of the current DD framework, balancing information gain with bi-level optimization complexity.
> > > * **The semantic anchor acts as an optimization stabilizer at extreme compression.** At SPC=2 and SPC=5, where data capacity is sufficient, both Clean-Attn and AH perform similarly. However, at the extreme limit of DD (SPC=1), the Clean reference heavily struggles with overfitting due to strict scarcity. In this harsh regime, the AH semantic trigger ('film') naturally guides the bi-level optimization trajectory. It anchors the gradients along the primary task features, allowing the model to maintain an optimal CTA of 89.29%.
> > > * **Backdoor robustness is independent of overfitting.** Regardless of whether the clean reference suffers from overfitting (SPC=1) or successfully mitigates it via data scaling (SPC ≥ 2), AH consistently achieves an ASR above 98%.
> > >
> > > **Conclusion:** While standard algorithmic regularizers catastrophically collapse the bi-level process, we empirically confirmed that expanding the SPC budget is a valid approach to mitigate clean reference overfitting in DD. The anomalous CTA boost observed in the AH setup at SPC=1 is a direct consequence of the semantic trigger stabilizing the extreme compression dynamics. Most importantly, explicit data-level mitigation does not wash out the backdoor, validating the robust structural threat of AH. We will incorporate these validations into the final manuscript.
> > >
> > > We sincerely hope these rigorous empirical results adequately address the concern regarding overfitting mitigation, and we remain entirely at your disposal should any final clarifications be needed.

---

### Decision · Program_Chairs · 2026-04-30

**Decision:**

Accept (regular)

**Comment:**

This paper introduces Attention Hijacking, a structural backdoor attack that targets attention-guided dataset distillation by manipulating distilled attention labels during the optimization process. The proposed method achieves strong attack success rates while preserving clean accuracy across multiple datasets and encoder-based transformer models, and several reviewers found the empirical study careful and convincing.

At the same time, reviewers pointed out that the current study is restricted to encoder-only architectures and classification settings, which limits the broader relevance of the attack to more recent generative-model pipelines. The Semantic Anchoring Hypothesis is also mainly supported through empirical observations, and the discussion of defenses remains relatively limited in scope. The author responses provided additional clarification on the intended threat scenario and discussed why extending the method to decoder-only models is nontrivial under existing distillation frameworks. While these explanations help contextualize the contribution, some questions about generality and defense coverage remain open. Overall, given the novelty of the attack perspective and the strength of the experimental results, I recommend a weak accept.